# Design, Synthesis, In Silico Studies and In Vitro Evaluation of New Indole- and/or Donepezil-like Hybrids as Multitarget-Directed Agents for Alzheimer’s Disease

**DOI:** 10.3390/ph16091194

**Published:** 2023-08-22

**Authors:** Violina T. Angelova, Borislav Georgiev, Tania Pencheva, Ilza Pajeva, Miroslav Rangelov, Nadezhda Todorova, Dimitrina Zheleva-Dimitrova, Elena Kalcheva-Yovkova, Iva V. Valkova, Nikolay Vassilev, Rositsa Mihaylova, Denitsa Stefanova, Boris Petrov, Yulian Voynikov, Virginia Tzankova

**Affiliations:** 1Department of Chemistry, Faculty of Pharmacy, Medical University of Sofia, 1000 Sofia, Bulgaria; ivalkova@pharmfac.mu-sofia.bg (I.V.V.); y_voynikov@pharmfac.mu-sofia.bg (Y.V.); 2Institute of Biodiversity and Ecosystem Research, Bulgarian Academy of Sciences, 1113 Sofia, Bulgaria; bobogeorgiev5@gmail.com (B.G.); nadeshda@abv.bg (N.T.); 3Institute of Biophysics and Biomedical Engineering, Bulgarian Academy of Sciences, 1113 Sofia, Bulgaria; tania.pencheva@biomed.bas.bg (T.P.); pajeva@biomed.bas.bg (I.P.); 4Institute of Organic Chemistry with Centre of Phytochemistry, Bulgarian Academy of Sciences, 1113 Sofia, Bulgaria; miroslav.rangelov@orgchm.bas.bg (M.R.); nikolay.vassilev@orgchm.bas.bg (N.V.); 5Department of Pharmacognosy, Faculty of Pharmacy, Medical University of Sofia, 1000 Sofia, Bulgaria; dzheleva@pharmfac.mu-sofia.bg; 6Faculty of Computer Systems and Techologies, Technical University–Sofia, 1000 Sofia, Bulgaria; elena@tu-sofia.bg; 7Department of Pharmacology, Pharmacotherapy and Toxicology, Faculty of Pharmacy, Medical University of Sofia, 1000 Sofia, Bulgaria; rositsa.a.mihaylova@gmail.com (R.M.); daluani@pharmfac.mu-sofia.bg (D.S.); bobi.stoyanov@abv.bg (B.P.); vtzankova@pharmfac.mu-sofia.bg (V.T.)

**Keywords:** AChE/BChE, Alzheimer’s disease, neurodegenerative disorders, antioxidants, donepezil, melatonin, neuroprotection, SH-SY5Y, Neuro-2a, BBB, molecular docking, MT1 and MT2

## Abstract

Alzheimer’s disease (AD) is considered a complex neurodegenerative condition which warrants the development of multitargeted drugs to tackle the key pathogenetic mechanisms of the disease. In this study, two novel series of melatonin- and donepezil-based hybrid molecules with hydrazone (**3a–r**) or sulfonyl hydrazone (**5a–l**) fragments were designed, synthesized, and evaluated as multifunctional ligands against AD-related neurodegenerative mechanisms. Two lead compounds (**3c** and **3d**) exhibited a well-balanced multifunctional profile, demonstrating intriguing acetylcholinesterase (AChE) inhibition, promising antioxidant activity assessed by DPPH, ABTS, and FRAP methods, as well as the inhibition of lipid peroxidation in the linoleic acid system. Compound **3n**, possessing two indole scaffolds, showed the highest activity against butyrylcholinesterase (BChE) and a high selectivity index (SI = 47.34), as well as a pronounced protective effect in H_2_O_2_-induced oxidative stress in SH-SY5Y cells. Moreover, compounds **3c**, **3d**, and **3n** showed low neurotoxicity against malignant neuroblastoma cell lines of human (SH-SY5Y) and murine (Neuro-2a) origin, as well as normal murine fibroblast cells (CCL-1) that indicate the in vitro biocompatibility of the experimental compounds. Furthermore, compounds **3c**, **3d**, and **3n** were capable of penetrating the blood–brain barrier (BBB) in the experimental PAMPA-BBB study. The molecular docking showed that compound **3c** could act as a ligand to both MT1 and MT2 receptors, as well as to AchE and BchE enzymes. Taken together, those results outline compounds **3c**, **3d**, and **3n** as promising prototypes in the search of innovative compounds for the treatment of AD-associated neurodegeneration with oxidative stress. This study demonstrates that hydrazone derivatives with melatonin and donepezil are appropriate for further development of new AChE/BChE inhibitory agents.

## 1. Introduction

Alzheimer’s disease is a progressive neurodegenerative disorder that primarily affects memory and cognitive functions [1,2]. The etiology of AD has not been fully elucidated; however, several pathological mechanisms have been recognized in the development of the disease: the deposition of amyloid β (Aβ) plaques, hyperphosphorylation of microtubule-associated protein tau (MAPT), oxidative stress, inflammation, impaired homeostasis of bio metals, reduction in acetylcholine (ACh) levels, neuronal loss, and dysfunction of the cholinergic system [3]. A key hallmark in AD pathogenesis, which can exacerbate its progression, is oxidative stress, especially in cortical and hippocampal tissues of the brain [4,5]. As a result, elevated lipid levels in the brain render it susceptible to the detrimental effects of oxidative stress; the appropriate application of antioxidants can slow down the progression of AD, consequently minimizing neuronal degeneration [6].

Current approaches in AD pharmacotherapy based on the cholinergic hypothesis aim to increase the level of ACh. This can be achieved by influencing the activity of the cholinesterase enzymes (ChE). There are two isoenzymes which play a key role in hydrolyzing the mediator: ACh–acetylcholinesterase (AChE) and butyrylcholinesterase (BChE). AChE activity is prominent in a healthy brain; however, in the case of AD, it remains unchanged or is slightly lowered, whereas BChE function may increase [7]. The vast majority of modern pharmaceuticals for the treatment of AD rely on AChE inhibition, however, newer approaches are oriented towards dual targeting of both enzymes, partly to more effectively restore the AchE balance, but also on the account that both enzymes take part in Aβ aggregation [4,8]. In addition, AchE can form an AchE-Aβ complex, which is more neurotoxic than Aβ alone [9]. Donepezil, rivastigmine, and galanthamine are the first-line treatment options used to enhance brain ACh levels in AD patients. The mechanisms by which AChE inhibitors block the catalytic action of the enzyme fall into three major categories: competitive inhibition, pseudo-irreversible inhibitions, and irreversible inhibition [10]. Donepezil is a selective, noncompetitive, and rapidly reversible inhibitor [11]; galanthamine acts as a reversible and competitive acetylcholinesterase inhibitor that also up-regulates the expression of nicotinic receptors [12]; rivastigmine is pseudo-irreversible and also inhibits BChE [7,13]. Tacrine, the first reversible dual AChE and BChE inhibitor used in AD therapy, has been withdrawn from the market due to its hepatotoxicity [14].

In light of this, researchers continue to make efforts in the design and development of small molecules with good activity towards AChE because of their favorable “drug-like” properties, i.e., low molecular mass and good pharmacokinetic profiles, but also continue the quest for multi-target-directed ligands (MTDLs) with low hepatotoxicity and a good safety profile [15,16].

MTDLs are organic scaffolds designed to target multiple biological pathways simultaneously with the goal of synergistic enhancement of the therapeutic efficacy [15]. Due to the complex pathogenesis of AD, MTDLs currently represent the most promising solution in AD therapy, and their design has generated considerable research interest [17,18].

Over the years, molecules containing both melatonin scaffold and a donepezil fragment have been demonstrated to possess strong inhibitory activity against acetylcholinesterase [19,20,21,22]. Melatonin holds promise as a potential therapeutic agent for developing new multitarget hybrids against neurodegeneration, including AD, because it can modulate the balance of Aβ production/clearance and mitigate Aβ neurotoxicity [23,24,25]. It is well known that melatonin modulates ADAM10, BACE1, PIN1, and GSK3 levels and reduces Aβ production; promotes Aβ clearance systems, such as glymphatic/lymphatic drainage, and blood–brain barrier (BBB) transportation and autophagy; acts on the PrPc/mGluR5/Fyn/Pyk2 and Ca^2+^/mitochondria pathways to ameliorate Aβ oligomer-induced neurotoxicity; and improves cognitive function and sleep quality in AD patients [23]. The neuromodulator melatonin synchronizes circadian rhythms and related physiological functions through the actions of two G-protein-coupled receptors: MT1 and MT2 [26]. In this context, the indole scaffold is also one of the preferable pharmacophores and is considered an essential mediator between the gut-brain axis due to its neuroprotective, anti-inflammatory, β-amyloid anti-aggregation and antioxidant activities [27]. Additionally, the molecules carrying melatonin scaffold and hydrazone or sulfonyl hydrazone moieties have been shown to exhibit significant antioxidant properties [28] as well as AChE inhibitory activity [29,30,31,32].

Therefore, the new derivatives designed in this work aimed to act as potential AChE and BChE inhibitors with significant additional pharmacological properties. With the aim to improve the antioxidant activity of the designed hybrid molecules, we synthesized a series of 30 target compounds by variation in indole and/or donepezil fragments, supported by the preliminary molecular docking results. Thus, we prepared three families of hydrazine–hydrazone-based compounds as well as sulfonyl hydrazone analogues. The effectiveness of the new compounds was predicted by using in silico approaches.

This study employed molecular docking, ADME/Tox computer prediction, and biological assays to determine the potential of the new hybrids for AChE/BChE inhibition and in vitro neuroprotection against oxidative stress on human dopaminergic neuroblastoma SH-SY5Y cells. The antioxidant activity of the hybrid molecules was assessed using various methods, including free radical scavenging (DPPH), cation radical scavenging activity (ABTS), investigation of lipid peroxidation activity using a β-carotene-linoleic acid assay and the FRAP method. Additionally, the effect of the compounds on specific brain cells, such as human and mouse normal and malignant cell lines (CCL-1, SH-SY5Y, NEURO-2A), was also studied because the biocompatibility of the newly synthesized compounds is an integral part of their characterization. Furthermore, the most perspective compounds identified as promising AChE and/or BChE inhibitors were tested by in vitro PAMPA assay to evaluate their ability to access the brain.

## 2. Results and Discussion

### 2.1. Chemistry and Design Strategy of the Multifunctional Donepezil–Melatonin Hybrids

In this study, we developed novel melatonin–donepezil hybrids by combining two fragments that possess complementary properties. The melatonin and/or indole scaffold, in addition to its above-mentioned neurogenic profile, could demonstrate antioxidant and neuroprotective features and could also interact with the AChE-PAS via pie–pie interactions (π–π or aromatic–aromatic are stacking interactions between aromatic ring systems of the ligand and of the residue side chain of the receptor). The protonable *N*-benzyl piperidine which is present in the well-known AChE inhibitor donepezil can interact with AChE-CAS through cation–pie interaction (electrostatic interactions between a cationic atom or a positively charged functional group and the p-electron cloud of the aromatic ring). The last feature of the designed compounds is the linker between the groups interacting with the PAS and CAS sites: the hydrazone and sulfonyl hydrazone fragments. It is known that the crystallographic structure of AChE includes a peripheral cationic site (PAS) at the entrance and a catalytic active site (CAS) at the bottom of the receptor cavity. Hence, inhibitors that bind to either site could inhibit the AChE.

The 30 target compounds from two series hydrazide–hydrazones **3a–r** and sulfonyl hydrazones **5a–l** were synthesized from appropriate aldehydes **1a–n** and hydrazides **2a–f** and/or sulfonyl hydrazides **4a–c** by a one-step synthesis [33], shown in Figure 1. Thus, the designed aroylhydrazone **3a–r** or sulfonyl hydrazone **5a–l** framework (Figure 1) was made in two parts: substituted indole or *N*-benzyl piperidine/*N*-benzyl pirrolidine, or substituted phenyl as a mainstay and substituted indole heterocycle at the fifth position as well as phenyl-substituted aromatic rings, connected to hydrazide or sulfonyl hydrazide moiety to intensify the desired pharmacophoric behavior with drug-like properties as well as antioxidant activities, cholinesterase inhibition, and neuroprotective effects.

The precise structure elucidation of all the synthesized scaffolds was carried out utilizing HREI-MS and NMR (^1^H- and ^13^C-NMR) spectroscopic tools. The spectral analyses were in accordance with the designed structures. (See the Appendix A, ^1^H NMR, ^13^C NMR and HRMS spectra, Appendix A). The ^1^H-NMR spectra of hydrazide–hydrazones (**3e–r**) showed single signals corresponding to resonances of azomethine protons (CH=N) at 7.27–7.67 ppm with the exception of compounds with donepezyl fragment **3a–d** which showed a doublet corresponding to the resonances of azomethine protons (CH=N) at 7.85–8.49 ppm. For the sulfonyl hydrazones (**5a–l**), the azomethine protons (CH=N) were in the range from 7.19 to 8.10 ppm. The NH protons were observed at 10.75–11.54 ppm, respectively. ^13^C-NMR spectra of **5a–g** exhibited resonances of azomethine (HC=N) carbons at 149.22 to 155.26 and hydrazide/hydrazone (C=O) carbons for the compounds **3a–r** at 165.28–175.01, respectively. 2D homonuclear correlation (COSY), DEPT-135 and 2D inverse detected heteronuclear (C–H) correlation (HMQC and HMBC) NMR experiments were used for the precise structure elucidation of all new compounds.

The stereochemistry of hydrazide–hydrazones **3a–r** was unambiguously confirmed with the help of cross-peak intensities observed in the 2D NOESY (nuclear Overhauser effect spectroscopy) spectrum. Although the four isomers were considered (Appendix A) [34,35] for the aroylhydrazones with indole scaffold, *E/Z* isomerization was generally not observed and the *Z* geometric isomers were absent. Only the ^1^H NMR spectrum of compound **3c** and **3o** taken in DMSO-d_6_ at 20 °C shows a 1:1 mixture of conformers. According to the confirmed nuclear Overhauser effect (NOE) between the methylidene proton and the NH proton, the most stable were the *E* isomers around the C=N double bond and the synperiplanar conformer around the amide O=C–N-N bond. Therefore, we concluded that a single *E* geometrical isomer was observed and the duplication pattern in the novel hydrazone derivatives to be due to the presence of syn/anti amide conformers in DMSO-d_6_. Additionally, for the purposes of structure elucidation, the NMR spectra of compound **3c** were measured at 363 K (compound of **3o** at 353 K) to achieve a fast exchange. After cooling down to 293 K, the ^1^H NMR spectra of **3a** and **3o** remained unchanged.

### 2.2. In Vitro AChE and BChE Inhibition Assays

The AChE and BChE inhibitory activity results of the synthesized compounds from two series, hydrazones **3a–r** and sulfonyl hydrazones **5a–l,** are provided in Table 1 and Table 2. To assess the selectivity of all the final synthesized compounds, the improved Ellman spectrophotometric method was employed, with certain modifications [36,37]. Donepezil and galanthamine were used as reference compounds. The results are presented by the IC_50_ as the mean of three measurements ± standard deviation. The enzyme inhibition percentage at 1 mM of the compounds is also given (Table 1 and Table 2).

None of the 30 tested compounds proved to be a more potent AChE inhibitor than the positive controls galanthamine and donepezil. The piperidinyl-containing derivatives (**3a–d** and **5a**) belonging to the first family showed relatively high IC_50_ values (76.51 ± 3.04, 19.84 ± 0.95, 10.76 ± 1.66, 9.77 ± 0.76 and 162.83 ± 10.73 μM, respectively). The derivative **3d** with a p-MeO benzene ring had the lowest IC_50_ (9.77 ± 0.76) in the family. The presence of the methoxy group on the appropriate position in the benzene ring (**3d**) lead to more of an increase in inhibitory activity than the *o-*, *p*-disubstituted benzene ring (**3b**) and indole containing hybrids **3a** and **3c**, increasing the selectivity index for the compound **3d** (SI = 20.19). Also, replacing the indole moiety in **3c** with a 2,4-dihydroxy benzene ring in compound **3b** caused a decrease in the AChE inhibitory activity. It is impressive that the introduction of a methylene group in the connecting hydrazone fragment in compound **3a**, compared to the indole-donepezil hybrid **3c** with the same moieties, lead to a loss of inhibitory activity towards AchE and an increase in activity towards the butyrylcholinesterase enzyme; the selective index is in favor of BChE (SI BChE = 1.35). The SAR, including molecular modelling studies, disclosed that the benzyl moiety provided interaction with CAS residues, while the indole moiety was oriented within PAS. On the other hand, compound **5a** with sulfonyl hydrazone linkage showed the best IC_50_ (162.83 ± 10.73 μM) from this family, indicating that the sulfonyl hydrazone fragment probably decreased the inhibitory activity.

Pyrrolidine-containing derivatives **3e–i** from the second family displayed highly variable results. Among them, the derivatives **3f** and **3h** exhibited moderate inhibitory activity with IC_50_ values of 549.73 ± 37.08 and 351.37 ± 17.93 μM, respectively. However, derivatives **3e**, **3g**, and **3i** showed significantly lower inhibitory activity and were practically inactive. It appeared that reducing the size of the piperidine ring to pyrrolidine, as well as the turn of the hydrazide hydrazone bond, resulted in a drastic decrease in inhibitory activity for both acetyl and butyrylcholinesterase. Comparing the compounds containing *p*-hydroxy groups and *m*-, *p*-dihydroxy groups in benzene rings (**3f** and **3h**) with *o*-, *p*- dihydroxy and *o-p-* trihydroxy benzene rings (**3e** and **3i**), it seemed that a combination of two hydroxyl groups at the ortho- and para-position assisted AChE inhibition. The presence of three hydroxyl groups in the *o-* and *para-* position of the benzene ring led to increased BChE inhibitory activity (SI BChE = 1.96).

Other *tert*-butyl-2-hydrazinyl-2-oxoethylcarbamate derivatives from the third family with different substitutions in the indole ring (**3j–m**) did not show significant inhibitory activity towards both AchE and BchE in the measured concentration, compared with the results obtained from H. Zhang et al., 2022 [38]. 

Finally, derivatives from the fourth family (**3n–r**) with different substitutions in the indole ring and hydrazide hydrazone linkage as well as their analogues with sulfonyl hydrazone linkage (**5b–l**) did not show satisfactory AChE inhibitory activity either, probably due to a lack of protonated nitrogen-containing moiety to ensure binding with CAS residues. Two sets of compounds inhibited BChE more than AChE: those containing a tosyl group (**5j, 5g, 5h**) and those with a *tert*-butyl-2-hydrazinyl-2-oxoethylcarbamate group (**3j–m**). All compounds containing a vanillin moiety (**3i, 3q,** and **3r**) showed very weak to no inhibition to both AChE and BChE. Surprisingly, the compound **3n** with two indole scaffolds showed the highest activity against BChE (IC_50_ = 21.12 ± 1.48 μM) and a high selectivity index (SI = 47.34) (Table 1). The difference in BChE inhibition between **3n** and another melatonin-containing derivatives might be due to the presence of a melatonin fragment and other indole rings in the molecule.

As can be seen, compounds **3c, 3f,** and **5a** did not show a remarkable selectivity for the AChE enzyme more than BChE and, thus, they can be considered non-selective cholinesterase inhibitors. From the above-mentioned results, it can be concluded that those compounds are non-selective AChE/BChE inhibitors with moderate activity against AChE.

### 2.3. Cytotoxicity of the Compounds

A series of MTT experiments were conducted against normal murine fibroblast cells (CCL-1) and malignant neuroblastoma cell lines of human (SH-SY5Y) and murine (Neuro-2a) origin to accommodate the in vitro biocompatibility of the experimental compounds. The results obtained from the cell viability assays are presented in Table 3.

The estimated half-inhibitory concentrations indicate a favorable cytotoxic profile of most compounds in both series with IC_50_ values ranging near or exceeding 100 μM in all three cellular test systems. However, the highest biocompatibility was observed in the hydrazide hydrazone series, where benzylpiperadine- (**3a–d**), pyrrolidine- (**3e**, **3g**, **3h**), *tert*-butyl- (**3j**, **3k**, **3l**) and vanillin- (**3r**) carrying representatives of all families were virtually devoid of cytotoxic activity (prevailing IC_50_ values were >300 and >800 μM, as seen from Table 3, many times exceeding the correspondent ones for the referent drug donepezil). In the alternative sulfonyl hydrazone series, the weakest effect on cell viability was again observed for the benzylpiperadine-substituted analogue **5a**, followed by **5d**, **5f**, **5k,** and **5l**.

The cytotoxicity profiles of the newly designed 30 hybrid compounds are in a good correlation with their antioxidant capacity, evaluated in several in vitro settings and presented in the section below. In a complimentary manner, the leading compounds with prominent free radical scavenging and cholinesterase inhibitory activity (**3a–d, 3h–l, 5a**) did not affect cell viability of neither normal nor malignant cell cultures. 

### 2.4. Determination of Antioxidant Activity

The antioxidant activity of the most perspective synthesized compounds was studied using three different spectrophotometric methods. The results are presented in Table 4 and Table 5.

### 2.5. DPPH Radical Scavenging Activity

The DPPH (2,2-diphenyl-1-picryl-hydrazyl-hydrate) free radical scavenging activity method is based on the neutralization of the DPPH radical by an antioxidant. Four compounds **5k**, **5j**, **5g**, and **3r** revealed the highest DPPH activity, with means from 45.17 ± 4.65% to 47.36 ± 2.75% (Table 4). Significant antioxidant activity was shown by the hydrazone compound **3r** (45.17 ± 4.65), characterized by the presence of three hydroxyl groups in the benzene ring and a vanillin moiety. The hydrazone-containing compound with a vanillin fragment and pyrrolidine scaffold, **3i** (6.56 ± 0.78), demonstrated noteworthy activity. The presence of multiple hydroxyl substituents attached to both aromatic rings created rich conjugated systems, making the hydrazones strong antioxidant agents. This ability allows them to effectively scavenge free radicals in their solutions by readily undergoing hydrogen radical abstraction (H), transforming into free radicals themselves. The present results are in agreement with the results obtained by other researchers [39,40] which showed that conjugated systems resulting from aromatic Schiff bases containing multi-hydroxyl groups have much higher antioxidant activities. Other hydrazone-containing compounds with moderate antioxidant activity better than BHT (butylated hydroxytoluene) were **3m, 3c,** and **3d**. The displacement of the hydrazone with sulfonyl hydrazone linkage produced a consistent increase in the activity, as seen for compounds **5g**, **5j,** and **5k** with 1-benzyl-1*H*-indol-3-yl or 5-(benzyloxy)-3-methyl-1*H*-indole fragments.

### 2.6. ABTS Radical Scavenging Assay

These compounds were also tested for their antioxidant activities by using the ABTS [2,2′-azinobis (3-ethylbenzothiazoline-6-sulfonic acid)] cation radical decolorization method (Table 4). The BHT [41] was used for comparison. The ABTS cation radical is produced by oxidation of the ABTS molecule with strong oxidizing agents, such as potassium persulfate in water. The produced greenish solution of this radical absorbs light at a wavelength of 734 nm. The advantage of this radical is the solubility in inorganic and organic solvents. Therefore, by using this assay, both hydrophilic and hydrophobic compounds can be tested. Another advantage of the ABTS assay is that the bulky compounds can approach this molecule to transfer electrons easier, compared to the DPPH molecule. It is well known that as a result of the polar effect of the substituents, the N-H bond in the structure of hydrazones and sulfonyl hydrazones is weakened, which increases the tendency of reacting with the ABTS molecule and more polar hydrazones were more amenable to react with ABTS, compared to less polar ones. In the ABTS assay, among the tested compounds, sulfonyl hydrazones exhibited higher cation radical scavenging activity. As shown in Table 4, most of the compounds demonstrated superior antioxidant activities than BHT and similar to melatonin, where hydrazine derivatives **3i, 3d,** and **5h** showed the lowest values of IC_50_, followed by sulfonyl hydrazones 5**j**, **5g**, **5k**, and hydrazones **3q**, **3n**, **3r**.

### 2.7. Ferric Reducing Antioxidant Power (FRAP)

The Ferric Reducing Antioxidant Power (FRAP) test measures the reduction in the complex of ferric ions (Fe^3+^)-ligand to the intensely blue ferrous complex (Fe^2+^) by means of antioxidants in acidic environments. Unlike other methods, the FRAP test is carried out in acidic pH conditions (pH = 3.6) to maintain solubility of iron complexes. Thus, the FRAP assay utilizes an SET (single electron transfer) mechanism to determine the capacity of an antioxidant in the reduction of an oxidant. With respect to the FRAP method, compound **3c** has the strongest activity, followed by **5a**, **5h**, donepezil, **3m** and **5j**.

The FRAP values of the tested compounds, as displayed in Table 1, showed a different trend than the DPPH radical scavenging results. Some of the compounds had lower activity in the DPPH assay but exhibited high activities in the FRAP assay. Compound **3c,** which was mildly active in the DPPH assay, showed the highest ferric reducing activity while compounds **5j** and **5g**, which were highly active in the DPPH assay, showed an unusually low FRAP value. The probable reason for these differences is that the reducing capacity of a sample in FRAP test is not directly related to its radical scavenging capability. Also, not all antioxidants reduce ferric ions at a fast rate.

In this study, according to the antioxidant activity assay results of the synthesized compounds, most of the hydrazones and sulfonyl hydrazones with indole and vanillin aromatic rings exhibited good antioxidant activity with the potential of forming free radicals by weakening the N-H bond in the hydrazone group.

### 2.8. Determination of Antioxidant Activity in Linoleic Acid System by the FTC Method

In the present study, the inhibition of lipid peroxidation of the synthesized compounds (10 mM) was determined using the FTC method in the linoleic acid system (Table 5). During lipid peroxidation, peroxides were formed and oxidized Fe^2+^ to Fe^3 +^. The Fe^3+^ ions formed a complex with SCN^–^, with an absorption maximum of 500 nm. Thus, a high absorbance was an indication of high peroxide formation during the emulsion incubation. The presence of compounds with antioxidant properties in the mixture decreased the linoleic acid oxidation and reduced absorption, respectively. The highest significant diminution was demonstrated by **3a** followed by **3c**, 5**h**, and **5j**. Moreover, most of the studied compounds demonstrated activity close to that of BHT, but inhibited lipid peroxidation stronger than the donepezil, melatonin, and control (Table 5).

The presented data of direct and indirect antioxidant activities as measured by the DPPH, ABTS, FRAP, and FTC methods (Table 4 and Table 5) demonstrate that aromatic hydrazones and sulfonyl hydrazones are promising antioxidant agents. The compound **3c**, which was a promising acetylcholinesterase inhibitor from all new derivatives, showed the best antioxidant activity in the three tested methods FRAP, ABTS, and FTC. As a future perspective, hydrazone **3c** will be evaluated in vivo in models of Alzheimer’s disease and melatonin deficiency as well as Aβ (1–42) aggregation.

Consequently, the prioritized structures (**3a–d**, **3n** and **5a**) were further subjected to an in vitro evaluation of their neuroprotective properties against H_2_O_2_-induced oxidative stress damage.

### 2.9. Neuroprotection against Oxidative Stress

The SH-SY5Y cell line is a commonly utilized human neuronal cell model that holds significant value in elucidating the mechanisms underlying neurotoxicity within neurodegenerative disorders [42,43]. It serves as an essential tool for investigating and comprehending the intricate processes involved in neurotoxicity associated with such diseases. To assess the potential neuroprotective properties of new compounds, an experimental setup was conducted where SH-SY5Y cells were subjected to pre-treatment with varying concentrations (0.1, 1, 5, 10, 25, 50 μM) for a duration of 90 min. Subsequently, the cells were exposed to 1 mM H_2_O_2_, resulting in a notable reduction in cell viability, confirming its toxicity (Figure 2). The mechanism of toxicity of H_2_O_2_ (hydrogen peroxide) in vitro involves oxidative stress and damage to cellular components. H_2_O_2_ can readily cross cell membranes and generate reactive oxygen species (ROS) through its spontaneous decomposition. ROS, including hydroxyl radicals and peroxyl radicals, can cause damage to DNA, proteins, and lipids, leading to cellular dysfunction and injury [44].

Therefore, we performed an investigation to assess the potential protective effects of the newly synthesized compounds using an in vitro model of hydrogen peroxide-induced oxidative stress.

The application of melatonin, a reference compound, in a pre-treatment regimen for a duration of 90 min exhibited remarkable benefits in enhancing the resilience of SH-SY5Y cells against H_2_O_2_-induced damage. This was evidenced by a substantial increase in cell viability compared to the H_2_O_2_ group, with percentages of 10%, 18%, 30%, 44%, and 52%, for melatonin concentrations of 1, 5, 10, 25, and 50 μM, respectively. A statistical analysis revealed significant differences (*p* < 0.05; *p* < 0.001) when compared to the H_2_O_2_ group. Likewise, the compound rasagiline exhibited a protective influence at concentrations of 5, 10, 25, and 50 μM, resulting in viability percentages of 14%, 28%, 43%, and 45%, respectively. The pre-incubation with compound **3n** effectively preserved the integrity of SH-SY5Y cells, indicating a potential neuroprotective effect. Notably, compound **3n** displayed a more pronounced protective effect when compared to the reference compound donepezil. Furthermore, at the highest concentration employed, it demonstrates the most significant neuroprotective activity among the three reference compounds, providing 60% protection (*p* < 0.001).

In summary, although none of the 30 tested compounds proved to be a more potent AChE inhibitor than donepezil than the compounds designed from Wang et al. [45] based on the fusion of donepezil and melatonin, our lead molecule **3c** manifests itself as a potential multifunctional agent with antioxidant activity better than donepezil–melatonin in the DPPH and FRAP methods, inhibited lipid peroxidation stronger than donepezil–melatonin and the control, showed better biocompatibility than donepezyl, and also displayed very low cytotoxic activity against normal murine fibroblast cells (CCL-1) and malignant neuroblastoma cell lines of human (SH-SY5Y) and murine (Neuro-2a). Additionally, the pharmacokinetic properties and the blood–brain barrier (BBB) permeability of compound 3c were favorable and suitable for further study in vivo. Additionally, the compound 3n with two indole scaffolds showed the highest activity against BChE (IC_50_ = 21.12 ± 1.48 μM), a high selectivity index (SI = 47.34), and effectively preserved the integrity of SH-SY5Y cells in the in vitro model of hydrogen peroxide-induced oxidative stress, indicating a more pronounced protective effect than the reference compound donepezil.

### 2.10. Molecular Docking of Human AChE and of Human BChE

The thirty hydrazide hydrazone and sulfonyl hydrazone derivatives were subjected to molecular docking in AChE and BChE crystallographic structures. Studies were conducted using the X-ray crystallographic structures of human AChE (PDB ID 4EY7) and human BChE (PDB ID 6QAA). Figure 3 presents the 2D protein–ligand interaction (PLIs) diagrams of the co-crystalized ligands in AChE (Figure 3a) and BChE (Figure 3b), respectively.

The PLI diagrams of the reference compounds clearly demonstrate that water molecules mediate some of the PLIs in the active sites of the enzymes. Thus, the applied docking protocol involves water (solvent) molecules. In addition, the PLIs outline several amino acid residues that participate in specific interactions with the ligands. These residues were taken into account when analyzing the PLIs of the studied hydrazide hydrazones and sulfonyl hydrazones derivatives.

Molecular docking studies of the investigated compounds in AChE and BChE were performed by applying a “triangle matcher” option as a placement method and London dG scoring function. Different docking protocols were explored to follow the effect of docking parameters on the docking results. The following docking protocols (DP) were applied depending on the type of the post-placement refinement and the final scoring methodologies: DP 1: rigid receptor/no final scoring; DP 2: rigid receptor/London dG final scoring; DP 3: rigid receptor/GBVI/WSA final scoring; DP 4: induced fit/no final scoring; DP 5: induced fit/London dG final scoring; DP 6: induced fit/GBVI/WSA final scoring. For both targets, very similar results were obtained between DP 1 and DP 4, DP 2 and DP 5, and between DP 3 and DP 6, respectively (Figure 4). The analysis of the obtained results shows that the rankings obtained with DP 1 and DP 4 (Figure 4), lower charts) are in best agreement with the experimentally reported activities (Table 1 and Table 2). For clarity, Figure 4a shows the obtained results from rigid receptor docking (DP 1, DP 2 and DP 3) in AChE and from induced fit docking (DP 4, DP 5 and DP 6), in BChE (Figure 4b).

The docking protocols have been validated by inspecting the binding energies and the RMSD values of the redocked reference ligand (donepezil). The reported docking scores of donepezil (Figure 4) correspond to the poses with the lowest RMSD (between 0.367 and 1.053 Å for docking protocols DP1 to DP6) illustrating the ability of the applied protocols to reproduce the experimental pose of the reference ligand in very similar orientations and conformations. The parallel runs resulted in the same docking scores and poses. In addition, the ranking of compounds according to their activities and docking scores show good correspondence (Figure 4), thus confirming the ability of the applied docking protocols to approximate the binding energies of the studied compounds.

As seen from Figure 4, hydrazide hydrazone derivatives (3* compounds) show more favorable binding energies in comparison with sulfonyl hydrazone derivatives (5* compounds). The lower charts in Figure 4 show separately the results obtained by DP 1, where the most active compounds (see Table 1 and Table 2) are presented in magenta. In the case of AChE, the most active compounds occupy the positions with highest binding energies following donepezil. In the case of BChE, this tendency is less pronounced; however, all active compounds are classified in the top half.

In addition to the analysis based on the docking scores (binding energies) (Figure 4), a visual inspection of the poses of the most active compounds, along with those of donepezil, were also performed to ensure that the poses with the lowest docking scores correspond to the most reasonable binding modes of the compounds (the best overlap to the reference compound, similar orientations of the interacting functional groups and atoms of the ligand). Figure 5 illustrates the best poses of donepezil and of the active compounds **3a**, **3b**, **3c**, and **3d** in the binding site of AChE. As seen, the most active compounds overlay well with donepezil and especially in the CAS region of the pocket.

Figure 6 represents the PLIs diagrams of the best poses after docking (except for compound **3a**, where the third scored pose of binding energy −53.91 kcal/mol towards −57.88 kcal/mol of the best scored pose, as shown in Figure 4) of the most active compounds in AChE performed with DP 1 chosen as the most reliable one after a thorough analysis of the obtained results.

As seen from Figure 3a, the co-crystalized AChE donepezil demonstrated six PLIs: one H-bond with Phe295, two H-bonds mediated by water with Asp74 and Tyr337, one arene-arene interaction with Trp86, one arene-H interaction with Tyr341 and one arene-H interaction mediated by water with Trp286. The donepezil, after docking (Figure 6e), repeated all the PLIs of the co-crystalized ligand, showing one newly appeared PLI—an arene-H interaction with Ser203. As seen from Figure 6a–d, the most active compounds **3a**, **3b**, **3c**, and **3d** repeated the aforementioned PLIs of donepezil after docking with Asp74, Trp86, Ser203, and Tyr337. Almost all of the shown compounds repeated the arene-H interaction with Trp86, except compound **3b**, where this interaction appeared directly with Trp286 without mediating water. Almost all of the shown compounds performed the H-bond with Phe295, except compound **3a**. Two out of four shown compounds (**3a** and **3d**) repeated the arene-H interaction with Tyr341. All four compounds demonstrated a newly appeared interaction of type arene-H with Phe338. The results obtained by the molecular docking studies are in agreement with the results shown by Alov et al., [46] demonstrating the interactions with Trp86, Trp286, and Phe295 as of high importance for AChE. The interaction with Trp286 is one of the most frequent intermolecular interactions between AChE and donepezil, therefore its engagement in the interaction of our ligands and AChE mimics the natural ligand interaction.

The binding modes of the compounds of series 5 were also analyzed on the example of the best ranked and most active sulfonyl hydrazone derivative **5a** (Appendix A). In contrast to hydrazide hydrazone derivatives, the preferable ionized form of the compound at physiological pH (73% of all possible forms) was with a polarized =N-N- group, potentially due to the neighboring -SO_2_ group (Appendix A). In addition, except the best one, the remaining binding poses showed an opposite orientation in the receptor pocket with the benzyl-piperadine part directed to the PAS region of the cavity. Appendix A illustrates these results on the example of the first two highest ranked poses of compound 5a. Although the best ranked pose has an orientation similar to that of donepezil, the other poses (with similar docking scores) had opposite orientations in the binding site. Thus, the lower activity values and docking scores of the compounds of series 5 can be related to the above discussed structural peculiarities; however, additional studies should prove these suggestions.

Figure 7 represents the PLIs of compounds **3a**, **3c**, **3d,** and **3n** after docking performed with DP 1, respectively, in AChE and BChE. The compounds **3a**, **3c**, and **3n** were chosen as the most active ones in BChE, while the compound **3d** was chosen instead of the next most active compound **3h**, due to its highest activity in AChE. The best poses of compounds **3c** and **3n** are shown, while Figure 5a depicts the second scored pose of compound **3a** with a binding energy −45.11 towards −46.88 of the best scored pose; Figure 6c depicts the fifth scored pose of compound 3d with a binding energy −40.34 towards −43.22 of the best scored pose.

As seen from Figure 7e, the best scored pose of donepezil after docking in BChE demonstrated water-mediated H-bonds with Asp70 and Thr120. In Figure 7a–d, all compounds **3a**, **3c**, **3d**, and **3n** repeated the H-bond with Asp70, while the compounds **3a** and **3c** repeated the PLI with Thr120 as well. In addition, compounds **3a**, **3d**, and **3n** demonstrated a H-bond mediated by water (except **3a**) with Ser287. Compounds **3a** and **3n** demonstrated a H-bond with Ile69, as well as an arene-H interaction with Trp231.

The above analysis illustrates that the most active compounds in the series have similar binding modes to that of donepezil and perform specific interactions with residues involved in donepezil binding. Thus, based on the binding energies and protein–ligand interactions, the donepezil-based derivatives appear to be promising lead structures as inhibitors of the studied cholinesterases.

### 2.11. Docking of Ligand Set in MT1 and MT2 Receptors 

The neuromodulator melatonin synchronizes circadian rhythms and related physiological functions through the actions of two G-protein-coupled receptors: MT1 and MT2. Because MLT exerts neuroprotective and anti-apoptotic effects by the activation of MT1 receptors—probably through the induction of the expression of several antioxidant enzymes and the modulation of mitochondrial function by MT1 receptors localized on mitochondria in mouse models of neurodegenerative disorders—we docked the most promising AchE and BchE inhibitors from the molecular docking results discussed above to an MT1and MT2 crystal structure, prioritizing structural fit and chemical novelty.

### 2.12. MT1 Receptor Docking Results

The pocket in the MT1 receptor is mostly isolated from the outside media and its widest dimension is in the plane, perpendicular to spirals forming channels. There also exists a narrow protuberance in the inside of the cell. A visualization of the pocket of the MT1 receptor (Appendix A), as well as detailed data of the docking results, are presented in the Appendix A.

To present interactions between receptors and our ligands at their best poses, we prepared interaction maps at which polar amino acids are depicted with pink, while lipophilic ones are in green (Figure 8), acidic amino acids are circled with red, while the basic ones are circled with blue. Side chain interactions are depicted with green arrows, and the interaction with the backbone is in blue, where the arrowhead points to the hydrogen bond acceptor. Exposure to the solvent is depicted with a blue halo around the ligand atoms and by blue circles around the amino acids of the receptor. The proximity contour of the pocket is depicted by a gray dotted line.

The best interaction with MT1 is by **3a** (Appendix A). The best interaction energy in this case is due to its best fit in the active site cavity without big sterical hindrances and its major term is hydrophobic energy. Unfortunately, there are no formed hydrogen bonds or relatively high electrostatic interactions. It is worth mentioning that in the very close vicinity to the ligand are some key amino acids, of which the unique configuration is characteristic of the active form of the receptor. The interaction of Asn162 (N162 4.60), according to Ballesteros–Weinstein nomenclature [47], with neighboring amino acids and, thus, the formation of a H-bond, leads to modulation and a reduction in the entrance of the ligand. His195 (H195 5.46) is well known to be crucial for the formation of the “longitudinal channel” for ligands in MT1 and its mutation is deleterious [48]. Tyr281 (Y281 7.39) with two adjacent residues also participating in forming smaller pockets in MT1, allowing more accommodation in the MT1 of different ligands than MT2, thus helping in subtype specificity between them.

The next of the best interacting ligands is **3c**, which has a low hydrophobic energy gain due to the less hydrophobic solvent-accessible surface based on the lack of one CH_2_ group. It forms two hydrogen bonds with the amide group of Gln181 (Q181 ECL2) and with Asn255 (N255 6.52) that will lead to increased selectivity of that ligand to the pocket of MT1. Ramelteon, a well known agonist used for insomnia treatment, forms hydrogen bonds with Gln181 [49]; mutational studies confirm its importance for the binding of natural ligand melatonin and receptor activation. Gln181, together with closely situated Phe179 (F179 ECL2), is positioned in the ECL2 (extra cellular loop) with part of the receptor forming a “lid” structure, participating in ligand recognition and positioning it right for further activation. It is recognized in mutation analysis that the replacement of Gln181 severely diminished the function of the MT1 receptor to its natural ligand, therefore highlighting its importance. What is also impressive is the formed hydrogen bond with Asn255, the amino acid, known to form the only significant polar interaction of melatonin in the MT1 receptor [50].

An analysis of the best 50 poses of our ligands inside the MT1 cavity shows that most of the ligands interact by surface contact with Phe179 followed by arene contact with Phe196 (F196 5.47), side chain hydrogen bond forming with Asn255 (as donor), hydrogen bond forming with the backbone of Gly104 (G104 3.29) (as donor), and interacting with the side chain of Gln181 as a hydrogen bond acceptor.

Phe179 and Gln181 as part of ECL2 and their role in receptor activation, as well as Asn255 interaction, are discussed above. Stauch et al. [51] stated that the ligands interact with MT1 “mainly by strong aromatic stacking with Phe179 and auxiliary hydrogen bonds with Asn162 and Gln181”. Phe196 (F196 5.47) is one of the three most frequent intermolecular interactions between the MT1 receptor and melatonin, therefore its engagement in the interaction with our ligands and MT1 mimics the natural ligand interaction [50].

### 2.13. MT2 Receptor Docking Results

The widest dimension of the active site pocket of MT2 is in the plane, perpendicular to the spirals forming the channel. A narrow protuberance is formed inside of the cell that is bigger compared to the one in MT1and highly hydrophilic. A visualization of the pocket of the MT2 receptor (Appendix A) as well as detailed data of the docking results (see Appendix A) are presented in the Appendix A. Interaction maps of the best ligands by their interaction energy are presented in Figure 9.

An analysis of the best 50 poses of our ligands inside the MT2 cavity (Appendix A) shows that most of the ligands interact by surface contact with Phe192 (F192 ECL2) followed by interactions with Gln194 (Q194 ECL2), of which the side chain acts as an acceptor of the hydrogen bond. In most cases, Phe209 (F209 5.47) interacts by its arene ring and Ala117 (A117 3.29) acts as a donor of the hydrogen bond with its backbone atoms. Tyr294 (Y294 7.39) interacts as an acceptor of the hydrogen bond with its phenolic group.

Tyr 294 (Y294 7.39) is one of the amino acid residues that, together with Tyr 298 (Y298 7.43) and Leu295 (L295 7.40), is the reason for the formation of larger pockets in the MT2 receptor, allowing the accommodation of bigger ligands. Therefore, Tyr 294 is one of the amino acids participating in the formation of molecular structure and is selective for MT2 agonists [48].

A number of direct and water-mediated hydrogen bonds with the polar side chains of Gln194 ECL2, Asn162 (N162 4.60), and Asn268 (N268 6.52) are reported for simulations of ligand binding specific to the MT2 receptor [52], and Asn268 (N268 6.52) and Leu295 (L295 7.40) are reported in expression studies as essential for natural ligand binding to MT2 [52]. Phe209, on the other hand, is participating in the formation of the MT2 receptor specifically to the hydrophobic subpocket [53].

Ala117 is found to be one of the bulkier amino acids that is different in MT2 in comparison to MT1 and, thus, interaction with them could aid in the larger and more specific accommodation of MT2ligands [54]. Phe 192 from ECL2 is known to form an “aromatic sandwich” in interactions with specific ligands for MT2, thus is involved in the agonist specificity of the MT2 receptor [55]. As can be seen in Appendix A, with the exception of **5a** and **3i**, the ligands with a clear preference towards one of the receptors prefer the MT2 receptor. Only two of the ligands, **3a** and **3c**, do not discriminate between MT1 and MT2 receptors, but bind strongly to both of them. These ligands are the best binders for the MT1 receptor and are in the first four of the best binders for MT2 receptors.

Since MLT inhibits Aβ generation and prevents their aggregation and formation into amyloid fibrils through its neuroprotective and antioxidant properties in the brain via the activation of MT1 and MT2 receptors, we hope that newly synthesized compounds, especially **3c**, similarly to MLT would mediate antifibrillogenic effects and prevent cells from β Amyloid (Aβ)-mediated toxicity, the pathogenic process underlying AD.

### 2.14. In Silico ADME/Tox Study

The synthesized compounds were subjected to in silico ADME screening using the SwissADME online tool. Specific molecular and physicochemical properties were analyzed in comparison to available drugs used to treat AD (Table 6 for the hydrazide hydrazones and Table 7 for the sulfonyl hydrazones). These included molecular weight (MW), number of heavy atoms, number of aromatic heavy atoms, number of rotatable bonds, hydrogen bond donors (HBD), hydrogen bond acceptors (HBA), topological polar surface area (TPSA), molar refractivity (MR), octanol/water ratio (WLOGP), solubility, BBB permeability, PgP substrate, and gastro-intestinal absorption (GIA).

The obtained results showed average molecular weight (MW) levels below 450 for all newly synthesized compounds, which is essential for drugs that need to cross into and affect the central nervous system (CNS). The TPSA values were in the range of 60 Å² to 112 Å², which also fell within the limit below 140 Å² for active absorption of the molecules, good permeability, and oral bioavailability. In all derivatives, the rotatable bonds were between 5 and 7, except for compound **3m** where there were 10 [56]. Most CNS affecting compounds have 5 or fewer rotatable bonds, and more than 10 have been shown in rats to reduce the oral bioavailability [56].

The formation of hydrogen bonds is primarily associated with the bonding of oxygen and nitrogen in molecules. Thus, the probability of the molecule entering the CNS increases when the sum of nitrogen and oxygen atoms is less than five. The analysis showed that the number of proton acceptors was less than six and the number of proton donors was less than three for both hydrazide hydrazones and sulfonyl hydrazones. 

The n-octanol/water ratio (logPo/w) is a key physicochemical parameter related to the lipophilicity of compounds. The biological activity of drugs is almost entirely due to their logP, and the rate at which they are metabolized is linearly related to logP. LogP values between two and four were recommended for drugs used to treat neurological diseases where higher log *p* values indicate a higher rate of brain penetration of compounds, as well as the opposite [57]. From the obtained data shown in Table 6, we found that compound **3m** (3.66) had the highest WLOGP value among the hydrazide hydrazones followed by **3n** (3.35), both values being under that of donepezil (3.83). Most of the sulfonyl hydrazones (Table 7) showed acceptable values less than 4, except for a few molecules with higher values: **5g** (5.39), **5j** (5.30), and **5d** (5.08). 

Molar refractivity (MR) is another important descriptor frequently used in QSAR studies in drug design. To increase the drug similarity, logP values were recommended to be in the range of −0.4 to +5.6 and for molar refractivity from 40 to 130. The values of the relevant descriptors for all compounds fell within these indicated ranges (Table 6 and Table 7).

The results of the studies on the solubility of the proposed newly synthesized compounds were also presented in Table 6 and Table 7 Most of the hydrazide hydrazones were classified as soluble, except **3a**, **3c,** and **3m** (Table 6) that were marked as moderately soluble. In the group of the sulfonyl hydrazones (Table 7), eight compounds were predicted to be soluble (**5a**, **5b**, **5c**, **5e**, **5f**, **5h**, **5i**, and **5l**) and the rest as moderately soluble, according to the classic ESOL model.

### 2.15. Drug-Likeness Properties

Lipinski’s rule is important for the development of new drugs because initial screening with this rule allows optimization of the chemical structure of promising molecules in the preliminary analysis. Lipinski’s rule is a heuristic approach to predict drug similarity and determines that molecules with MM > 500, logP > 5, proton donors >5, and proton acceptors >10 have both poor absorption and permeability. From Table 6 and Table 7, it can be seen that the molecular weights of the new compounds were within the range 294.7–419.5 Da. The logP values of most of the hydrazones were smaller than 5 (in the range of 2.49–4.67), except for compounds **5d** (5.03), **5g** (5.39), and **5j** (5.30). The number of groups that accepted hydrogen atoms (n-ON) was less than 10, and the groups that donated hydrogen atoms (n-OHNH) were less than 5, which were within Lipinski’s rules.

### 2.16. In Silico Prediction of BBB Permeability

Drugs that target the CNS must first cross the blood–brain barrier (BBB). The inability of therapeutic molecules to penetrate the BBB is a key obstacle for CNS drug candidates and needs to be addressed quickly in the drug development process; predicting the BBB permeability of new CNS drugs is therefore crucial [54]. A SwissADME webserver was used to predict the BBB permeability [56]. The study confirmed that all compounds can penetrate the BBB, and this facilitates their action inside the CNS for treating AD. From the prediction results obtained (Table 6 and Table 7), it can be seen that only some of the proposed compounds meet the requirements for possible transfer to the brain: **3a**, **3c**, **3d**, **3e**, **3i**, **3o**, **3p**, **5a**, **5f**, and **5l**. Most of the hydrazide hydrazone structures were predicted to have the ability to act as a substrate for P-gp except compounds **3d**, **3l**, **3m**, and **3o–3r**. Unlike hydrazide hydrazones, none of the sulfonyl hydrazones showed properties of a P-gp substrate. The predicted GI data of the 30 compounds and the two control drugs are also presented in Table 6 and Table 7. All of them showed a high degree of GI absorption.

### 2.17. Predicted Toxicity

The results from the toxicity prediction experiments performed over the proposed hydrazide and sulfonyl hydrazones are presented in Table 8. The predicted values of the median lethal dose (LD_50_), the toxicity class, the logP value, and the main toxicological pathways and endpoints were calculated and presented along with the corresponding mean value of the probability (Prob). The predictive results indicated that almost all compounds are defined as Inactive with regard to hepatotoxicity, mutagenicity, immunotoxicity, and cytotoxicity. Weak activity (in blue) with a relatively low level of probability was established for compounds **3o** and **3r**, similar to the control drug donepezil. In terms of carcinogenicity, most compounds were determined to be weakly active (in blue) with low probability levels between 0.50 and 0.63. Only one of the tested compounds, **3r,** showed high activity (in red) in the immunotoxicity study with a probability level of 0.94, as well as donepezil itself (0.95). 

### 2.18. BBB Permeability by In Vitro PAMPA Test

Compounds **3a**, **3b**, **3c**, **3m**, **3n**, and **5a** were considered as promising CNS-acting agents based on their AChE and/or BChE inhibition potencies, selectivity, and favorable cytotoxicity profiles. At the same time, they needed to fulfill the requirement for good BBB permeation in order to achieve sufficient brain exposure. Although transport across the BBB is quite a complex phenomenon, passive diffusion is still the primary route for exogenous substances to access that target [58]. So, the selected compounds were subjected to an in vitro PAMPA-BBB assay, relying on the capacity of the method in distinguishing the candidates with good BBB passive permeabilities [59]. The results (Table 9) showed that five amongst the six tested compounds had a high BBB permeability of logPe < 5, and only 3b was moderately permeable with logPe  =  5.873.

Our experimental PAMPA-BBB study was accompanied by in silico calculations of the physicochemical properties responsible for an optimal BBB passage [37]. For the purposes of the present discussion, the following parameters were calculated by ACD/logD software v. 9.08, ACD Inc., Benton, AR, Canada: molecular weight (Mw), the most basic pKa,_MB_ value, logP, pH dependent octanol/water distribution coefficient calculated at pH 7.4 (logD_7.4_), polar surface area (PSA), number of free rotatable bonds (FRB), counts of hydrogen bond donors (HBD), and hydrogen bond acceptors (HBA) (Table 7).

All compounds except **3m** fulfill Lipinski’s rule of five, which is an indication of good gastro-intestinal permeability. Because of the anatomical and functional specificities of the BBB, more stringent criteria are applied to compounds aimed at action in the brain. The molecular weights of tested compounds are around the upper limit of 400 g/mol, proposed as a threshold for good BBB permeability [60]. With logP values between 3.16 and 5.05, logD_7.4_ in the range 2.56–5.05, 5–8 free rotatable bonds (FRB), 1–3 H-bond donors (HBD) and 5–7 H-bond acceptors (HBA), most of the compounds meet the requirements for the given properties and agreed well with the PAMPA-BBB experimental results [61,62].

Compounds **3a** and **3c** are moderate bases, **3m** is a week acid with pKa 10.85, **3b**, **3n**, and **5a** are amphoteric. Compounds **3a**, **3b**, **3c**, and **5a** are partially ionized at physiological pH 7.4 which is reflected by the difference between logP and logD_7.4_ values, while **3m** and **3n** exist mainly in a neutral state. Only **3c** and **5a** satisfy the polarity restriction criterion (PSA < 79 Å^2.^) [61]. Compounds **3a**, **3c**, and **5a** were also suggested as capable of escaping the P-gp mediated-efflux in the brain. The sum of nitrogen and oxygen atoms in their molecules is five (counted from the structure) and only slightly exceeds the proposed cut-off of four, but they completely obey the other two requirements of the rule of four, namely pKa,_MB_ < 8 and and MW < 400 [63]. The moderate BBB permeability of **3b** is undoubtedly attributed to its higher polarity and more extensive ionization, about 80% at pH 7.4.

From the viewpoint of the lead discovery process, compounds **3c** and **5a** were considered as the most prospective for further optimization since they most closely cover the rule of three recommendations [64].

## 3. Materials and Methods

### 3.1. Chemistry

All solvents, chemicals, and reagents were obtained commercially and used without purification. Thin layer chromatography (TLC) was used to monitor the reactions. All melting points were determined in open glass capillaries and are uncorrected. ^1^H spectra were recorded using DMSO-d_6_ as a solvent on a Brucker Advance III 600 MHz spectrometer with tetramethylsilane as an internal standard. Chemical shifts (δ) are given in parts per million (ppm). High resolution mass spectrometry (HRMS) analysis was performed with the use of Agilent Accurate-Mass Q-TOF LC/MS G6520B system with dual electrospray (DESI) source (Agilent Technologies, Santa Clara, CA, USA). Microplate reader EZ Read 800, Biochrom and Shimatzu 1203 UV-VIS spectrophotometer (Japan) were used for antioxidant activity. All determinations were performed in triplicate (n = 3).

### 3.2. General Procedure for the Synthesis of Compounds ***3a–r***

The solution of 20 mmol of the corresponding carbonyl compounds (**1a–n**) in 10 mL of absolute ethanol was mixed with a hot solution of 20 mmol (60 °C) aroylhydrazide (**2a–f**) in 10 mL of absolute ethanol and stirred for 1–8 h. The obtained crystalline precipitates were filtered, washed with ethanol-ether, recrystallized from ethanol.

***N’-[(E)-(1-benzylpiperidin-4-yl)methylidene]-2-(1H-indol-3-yl)acetohydrazide,* 3a** Yield: 68%; m.p. 167–168 °C. ^1^H NMR (400 MHz, DMSO-d_6_): 1:0.83 mixture of conformers; signals for major *synperiplanar* conformer around the amide bond: δ = 1.42–1.53 (m, 2H, CH2), 1.67–1.78 (m, 2H, CH2), 1.94–2.04 (m, 2H, CH2), 2.17–2.24 (m, 1H, H-4′), 2.77–2.82 (m, 2H, CH2), 3.46 (s, 2H, CH2), 3.90 (s, 2H, CH2), 6.98 (ddd, J = 1.00, 7.0, 8.0 Hz, 1H, H-5), 7.05 (ddd, J = 1.0, 7.0, 5.0 Hz, 1H, H-6), 7.17 (d, J = 2.3 Hz, 1H, H-2), 7.22–7.33 (m, 10H, H-2″, H-3″, H-4″, H-5″, and H-6″), 7.35 (d, J = 7.0 Hz, 1H, H-7), 7.44 (d, J = 5.3 Hz, 1H, CH), 7.53 (d, J = 8.2 Hz, 1H, H-4), 10.84 (s, 1H, NH), 10.85 (bs, 1H, NH); resolved signals for minor *antiperiplanar* conformer around the amide bond: 3.45 (s, 2H, CH2), 3.53 (s, 2H, CH2), 6.96 (ddd, J = 1.0, 7.0, 8.0 Hz, 1H, H-5), 7.07 (ddd, J = 1.0, 7.0, 5.0 Hz, 1H, H-6), 7.20 (d, J = 2.3 Hz, 1H, H-2), 10.89 (bs, 1H, NH), 11.06 (s, 1H, NH); ^13^C NMR (100 MHz, DMSO-d6): signals for major *synperiplanar* conformer around the amide bond: δ = 28.85 (CH2), 29.06 (CH2), 38.10 (C-4′), 52.45 (CH2), 62.39 (CH2), 108.22 (C-3), 111.24 (C-7), 118.19 (C-5), 118.65 (C-4), 120.84 (C-6), 120.95 (C-5), 123.83 (C-2), 126.80 (C-4″), 127.14 (C-4a), 127.39 (C-4a), 128.11 (C-3″ and C-5″), 128.73 (C-2″ and C-6″), 135.95 (C-7a), 138.51 (C-1″), 149.21 (CH), 166.66 (C=0); resolved signals for minor *antiperiplanar* conformer around the amide bond: 152.96 (CH), 172.20 (C=O). HRMS (ESI) *m*/*z*: calcd: [M+H]^+^ 375.217938. Found: [M+H]^+^ 375.2178.

***N’-[(E)-(1-benzylpiperidin-4-yl)methylidene]-2,4-dihydroxybenzohydrazide,* 3b** Yield: 59%; m.p. 201–203 °C. ^1^H NMR (400 MHz, DMSO-d_6_): δ = 1.46 (dd, J = 3.1, 11.7, 1H) and 1.52 (dd, J = 3.1, 11.7 Hz, 1H, CH_2_), 1.74 (dd, J = 2.7, 13.5 Hz, 2H, CH_2_), 2.00–2.05 (m, 2H, CH_2_), 2.22–2.34 (m, 1H, H-4′), 2.82 (d, J = 11.4 Hz, 2H, CH_2_), 3.48 (s, 2H, CH_2_), 6.27 (d, J = 2.3 Hz, 1H, H-3), 6.32 (dd, J = 2.3, 8.7 Hz, 1H, H-5), 7.23–7.35 (m, 5H, Ar), 7.67 (d, J = 5.0 Hz, 1H, CH), 7.72 (d, J = 8.7 Hz, 1H, H-6), 10.17 (bs, 1H, OH), 11.32 (bs, 1H, NH), 12.44 (bs, 1H, OH). ^13^C NMR (100 MHz, DMSO-d_6_): δ = 29.00 (C-3′ and C-5′), 38.53 (C-4′), 52.42 (C-2′ and C-6′), 62.36 (CH_2_), 102.81 (C-3), 105.87 (C-1), 107.19 (C-5), 126.86 (C-4″), 128.13 (C-3″ and C-5″), 128.78 (C-2′’ and C-6″), 129.35 (C-6), 138.41 (C-1″), 155.26 (CH), 162.49 and 162.53 (C-2 and C-4), 165.44 (C=O). HRMS (ESI) *m*/*z*: calcd: [M+H]+ 354.181218. Found: [M+H]+ 354.1811.

***N’-[(E)-(1-benzylpiperidin-4-yl)methylidene]-1H-indole-3-carbohydrazide,* 3c** Yield: 72%; m.p. 249–250 °C. ^1^H NMR (400 MHz, DMSO-d_6_, 363K): δ = 1.51 (dd, J = 3.7, 11.3 Hz, 1H, CH_2_), 1.57 (dd, J = 3.7, 11.1 Hz, 1H, CH_2_), 1.79 (dd, J = 3.4, 12.9 Hz, 2H, CH_2_), 2.09 (dt, J = 2.6, 11.3 Hz, 2H, CH_2_), 2.23–2.32 (m, 1H, H-4′), 2.82 (td, J = 3.5, 11.7 Hz, 1H, CH_2_), 3.49 (s, 2H, CH_2_), 7.12 (dt, J = 1.2, 10.9 Hz, 1H, H-5), 7.16 (dt, J = 1.4, 7.5 Hz, 1H, H-6), 7.21–7.26 (m, 1H, H-4″), 7.29–7.32 (m, 4H, C-3″, C-5″, C-2″, and C-6″), 7.44 (d, J = 7.9 Hz, 1H, H-7), 7.55 (d, J = 4.8 Hz, 1H, CH), 8.12 (bs, 1H, H-2), 8.17 (d, J = 7.3 Hz, 1H, H-4), 10.53 (bs, 1H, NH), 11.36 (bs, 1H, NH). ^13^C NMR (100 MHz, DMSO-d_6_, 363K): δ = 29.01 (C-3′ and C-5′), 37.97 (C-4′), 52.19 (C-2′ and C-6′), 62.11 (CH_2_), 108.42 (C-3), 111.44 (C-7), 120.21 (C-5), 120.95 (C-4), 121.73 (C-6), 126.43 (C-4″), 126.63 (C-4a), 127.74 (C-3″ and C-3″), 128.44 (C-2″ and C-6″), 129.27 (C-2), 135.69 (C-7a), 138.34 (C-1″), 150.89 (CH), 162.35 (C=O). HRMS (ESI) *m*/*z*: calcd: [M+H]+ 361.202288. Found: [M+H]+ 361.2021.

***N’-[(E)-(1-benzylpiperidin-4-yl)methylidene]-4-methoxybenzohydrazide,* 3d** Yield: 82%; m.p. 138–140 °C. ^1^H NMR (600 MHz, DMSO-d_6_): δ = 1.49 (q, J = 10.9 Hz, 2H) and 1.74 (d, J = 11.6 Hz, 2H, H-2′, and H6′), 2.01 (t, J = 11.0 Hz, 2H) and 2.82 (d, J = 11.2 Hz, 2H, H-3′, and H-5′), 2.25 (d, J = 4.7 Hz, 1H, H-1′), 3.47 (s, 2H, CH_2_), 3.82 (s, 3H, OCH_3_), 7.02 (d, J = 8.6 Hz, 2H, H-3, and H-5), 7.25 (tt, J = 2.0, 6.7 Hz, 1H, H-4″), 7.30–7.34 (m, 4H, H-2″, H-3″, H-5″, and H-6″), 7.67 (d, J = 5.1 Hz, 1H, CH), 7.84 (d, J = 8.8 Hz, 2H, H-2, and H-6), 11.28 (s, 1H, NH). ^13^C NMR (151 MHz, DMSO-d_6_): δ = 29.14 (C-2′ and C-6′), 38.57 (C-1′), 52.48 (C-3′ and C-5′), 55.37 (OCH_3_), 62.42 (CH_2_), 113.60 (C-3 and C-5), 125.58 (C-1), 126.82 (C-4″), 128.12 (C-3″ and C-5″), 128.76 (C-2″ and C-6″), 129.34 (C-2 and C-6), 138.51 (C-1″), 154.21 (CH), 161.80 (C-4), 162.23 (C=O). HRMS (ESI) *m*/*z*: calcd: [M+H]+ 352.201953. Found: [M+H]+ 352.20115.

***1-benzyl-N’-[(E)-(4-hydroxyphenyl)methylidene]pyrrolidine-3-carbohydrazide,* 3e** Yield: 84%; m.p. 240–241 °C. ^1^H NMR (600 MHz, DMSO-d_6_): 1:0.77 mixture of conformers; signals for major *synperiplanar* conformer around the amide bond: δ = 1.94–2.03 (m, 2H, H-4′), 2.40–2.42 (m, 2H, H-5′), 2.62–2.69 (m, 1H, 1/2H-2′), 2.88–2.91 (m, 1H, 1/2H-2′), 3.55–3.62 (m, 2H, CH_2_), 3.61–3.66 (m, 1H, H-3′), 6.79 (d, J = 8.6 Hz, 2H, H-3, and H-5), 7.22–7.26 (m, 1H, H-4″), 7.30–7.33 (m, 4H, H-2″, H-3″, H-5″, and H-6″), 7.44 (d, J = 8.7 Hz, 2H, H-2, and H-6), 7.85 (s, 1H, CH), 9.84 (bs, 1H, OH), 11.05 (s, 1H, NH); resolved signals for minor *antiperiplanar* conformer around the amide bond: 8.04 (s, 1H, CH), 9.87 (bs, 1H, OH), 11.10 (s, 1H, NH). ^13^C NMR (151 MHz, DMSO-d_6_): signals for major *synperiplanar* conformer around the amide bond: δ = 26.95 (C-4′), 41.47 (C-3′), 53.58 (C-5′), 56.43 (C-2′), 59.30 (CH_2_), 115.65 (C-3 and C-5), 125.37 (C-1), 126.75 (C-4″), 128.11 (C-2″ and C-6″), 128.28 (C-2 and C-6), 128.65 (C-3″ and C-5″), 139.16 (C-1″), 142.74 (CH), 158.99 (C-4), 175.01 (C=O); resolved signals for minor *antiperiplanar* conformer around the amide bond: 146.34 (CH), 159.22 (C-4), 169.78 (C=O). HRMS (ESI) *m*/*z*: calcd: [M+H]+ 324.170653. Found: [M+H]+ 324,1705.

***1-benzyl-N’-[(E)-(2,4-dihydroxyphenyl)methylidene]pyrrolidine-3-carbohydrazide,* 3f** Yield: 81%; m.p. 224–225 °C. ^1^H NMR (600 MHz, DMSO-d_6_): δ = 1:0.39 mixture of conformers; signals for major *synperiplanar* conformer around the amide bond: δ = 1.92–2.04 (m, 2H, H-4′), 2.41–2.48 (m, 2H, H-5′), 2.66–2.69 (m, 1H, 1/2H-2′), 2.82–2.94 (m, 1H, 1/2H-2′), 3.52–3.57 (m, 1H, H-3′), 3.57–3.61 (m, 2H, CH_2_), 6.29 (d, J = 2.3 Hz, 1H, H-3), 6.33 (dd, J = 8.4, 2.3 Hz, 1H, H-5), 7.25 (d, J = 8.5 Hz, 1H, H-6), 7.29–7.33 (m, 5H, Ar), 8.21 (s, 1H, CH), 9.91 (bs, 1H, OH), 11.32 (s, 1H, OH), 11.36 (s, 1H, NH); resolved signals for minor *antiperiplanar* conformer around the amide bond: 8.11 (s, 1H, CH), 9.79 (bs, 1H, OH), 10.11 (bs, 1H, OH), 11.04 (s, 1H, NH). ^13^C NMR (151 MHz, DMSO-d_6_): signals for major *synperiplanar* conformer around the amide bond: δ = 27.47 (C-4′), 41.23 (C-3′), 53.52 (C-5′), 56.83 (C-2′), 59.17 (CH_2_), 102.58 (C-3), 107.57 (C-5), 110.41 (C-1), 126.81 (C-4′’), 128.14 (C-2′’ and C-6′’), 128.51 (C-3′’ and C-5′’), 131.16 (C-6), 139.17 (C-1′’), 147.58 (CH), 159.27 (C-2), 160.52 (C-4), 169.58 (C=O); resolved signals for minor *antiperiplanar* conformer around the amide bond: 141.95 (CH), 157.97 (C-2), 160.22 (C-4), 174.44 (C=O). HRMS (ESI) *m*/*z*: calcd: [M+H]+ 340.165568. Found: [M+H]+ 340.1655.

***1-benzyl-N’-[(E)-(3,4-dihydroxyphenyl)methylidene]pyrrolidine-3-carbohydrazide,* 3g** Yield: 72%; m.p. 199–201 °C. ^1^H NMR (600 MHz, DMSO-d_6_): 1:0.65 mixture of conformers; signals for major *synperiplanar* conformer around the amide bond: δ = 2.57 (m, J = 8.3 Hz, 2H, H-4′), 2.63 (d, J = 8.45 Hz, 2H, H-5′), 3.32–3.37 (m, 1H, H-2′), 3.57 (t, J = 9.5 Hz, 1H, H-2′), 3.91 (dq, J = 6.2, 8.6 Hz, 1H, H-3′), 4.35 (d, J = 15.0 Hz, 1H) and 4.46 (d, J = 15.0 Hz, 1H, CH_2_), 6.75 (dd, J = 0.6, 7.7 Hz, 1H, H-5), 6.89 (dd, J = 1.2, 8.3 Hz, 1H, H-6), 7.09 (s, 1H, H-2), 7.23–7.25 (m, 2H, H-2″ and H-6″), 7.27–7.30 (m, 1H, H-4″), 7.34–7.37 (m, 2H, H-3″ and H-5″), 7.80 (s, 1H, CH), 9.15 (bs, 1H, OH), 9.39 (bs, 1H, OH), 11.23 (s, 1H, NH); resolved signals for minor *antiperiplanar* conformer around the amide bond: 7.96 (s, 1H, CH), 9.24 (bs, 1H, OH), 9.37 (bs, 1H, OH), 11.26 (s, 1H, NH). ^13^C NMR (151 MHz, DMSO-d_6_): signals for major *synperiplanar* conformer around the amide bond: δ = 33.54 (C-3′), 35.41 (C-4′), 45.79 (CH_2_), 48.98 (C-5′), 49.37 (C-2′), 113.21 (C-2), 116.05 (C-5), 120.37 (C-6), 126.02 (C-1), 127.73 (C-4″), 128.05 (C-2″ and C-6″), 129.06 (C-3″ and C-5″), 137.24 (C-1″), 144.49 (CH), 146.11 (C-3), 148.17 (C-4), 172.83 (C=O); resolved signals for minor *antiperiplanar* conformer around the amide bond: 147.81 (CH), 146.15 (C-3), 148.43 (C-4), 173.81 (C=O). HRMS (ESI) *m*/*z*: calcd: [M+H]+ 340.165568. Found: [M+H]+ 340.1646.

***1-benzyl-N’-[(E)-(2,4,6-trihydroxyphenyl)methylidene]pyrrolidine-3-carbohydrazide,* 3h** Yield: 65%; m.p. 215–217 °C. ^1^H NMR (600 MHz, DMSO-d_6_): 1:0.19 mixture of conformers; signals for major *synperiplanar* conformer around the amide bond: δ = 1.91–2.03 (m, 2H, H-4′), 2.41–2.47 (m, 2H, H-5′), 2.66–2.69 (m, 1H, H-2′), 2.81–2.85 (m, 1H, H-2′), 2.85–2.91 (m, 1H, H-3′), 3.57 (d, J = 12.8 Hz, 1H) and 3.60 (d, J = 13.0 Hz, 1H, CH_2_), 5.81 (s, 2H, H-3 and H-5), 7.23–7.27 (m, 1H, H-4″), 7.30–7.33 (m, 4H, H-2″, H-3″, H-5″ and H-6″), 8.49 (s, 1H, CH), 9.77 (bs, 1H, OH), 10.95 (bs, 2H, OH), 11.33 (s, 1H, NH); resolved signals for minor *antiperiplanar* conformer around the amide bond: 8.37 (s, 1H, CH), 9.80 (bs, 1H, OH), 10.51 (bs, 2H, OH), 11.12 (s, 1H, NH). ^13^C NMR (151 MHz, DMSO-d_6_): signals for major *synperiplanar* conformer around the amide bond: δ = 27.47 (C-3′), 41.19 (CH_2_), 53.50 (C-4′), 56.81 (C-5′), 59.16 (C-2′), 94.29 (C-3 and C-5), 98.80 (C-1), 126.81 (C-4″), 128.14 (C-2″ and C-6″), 128.51 (C-3″ and C-5″), 139.12 (C-1″), 145.15 (CH), 159.48 (C-2 and C-6), 161.34 (C-4), 169.21 (C=O). HRMS (ESI) *m*/*z*: calcd: [M+H]+ 356.160483. Found: [M+H]+ 356.1604.

***tert-butyl(2-{(2E)-2-[(1-benzyl-1H-indol-3-yl)methylidene]hydrazinyl}-2-oxoethyl)carbamate,* 3i**^1^H NMR, ^13^C NMR and HRMS spectra of compound **3i** were published elsewhere [33].

***tert-butyl(2-{(2E)-2-[(5-methoxy-1-methyl-1H-indol-3-yl)methylidene]hydrazinyl}-2-oxoethyl)carbamate,* 3j** Yield: 48%; m.p. 203–206 °C. ^1^H NMR (600 MHz, DMSO-d_6_): 1:0.28 mixture of conformers; signals for major *synperiplanar* conformer around the amide bond: δ = 1.41 (s, 9H, CH_3_), 3.79 (s, 3H, OCH_3_), 4.16 (d, J = 6.0 Hz, 2H, CH_2_), 6.76 (t, J = 5.9 Hz, 1H, H-4), 6.85 (dd, J = 2.6, 8.8 Hz, 1H, H-6), 7.34 (d, J = 8.8 Hz, 1H, H-7), 7.62 (d, J = 2.4 Hz, 1H, H-4), 7.72 (d, J = 2.9 Hz, 1H, C-2), 8.14 (s, 1H, CH), 11.00 (s, 1H, NH), 11.40 (bs, 1H, NH); resolved signals for minor *antiperiplanar* conformer around the amide bond: 8.35 (s, 1H, CH), 11.09 (s, 1H, NH), 11.40 (bs, 1H, NH). ^13^C NMR (151 MHz, DMSO-d_6_): signals for major *synperiplanar* conformer around the amide bond: δ = 28.22 (CH_3_), 41.39 (CH_2_), 55.05 (OCH_3_), 77.88 (OC), 103.48 (C-4), 111.10 (C-3), 112.16 (C-7), 112.53 (C-6), 124.63 (C-3a), 130.50 (C-2), 131.93 (C-7a), 140.68 (CH), 154.37 (C-5), 155.89 (OC=O), 169.74 (C=O). HRMS (ESI) *m*/*z*: calcd: [M+H]+ 347.171382. Found: [M+H]+ 347.1712.

***tert-butyl(2-{(2E)-2-[(5-methoxy-1H-indol-3-yl)methylidene]hydrazinyl}-2-oxoethyl)carbamate,* 3k** Yield: 83%; m.p. 197–199 °C. ^1^H NMR (600 MHz, DMSO-d_6_): 1:0.33 mixture of conformers; signals for major *synperiplanar* conformer around the amide bond: δ = 1.41 (s, 9H, CH_3_), 3.78 (s, 3H, NCH_3_), 3.80 (s, 3H, OCH_3_), 4.16 (d, J = 5.9 Hz, 2H, CH_2_), 6.76 (t, J = 5.9 Hz, 1H, NH), 6.92 (dd, J = 2.6, 8.8 Hz, 1H, H-6), 7.41 (d, J = 8.8 Hz, 1H, H-7), 7.62 (d, J = 2.5 Hz, 1H, H-4), 7.71 (s, 1H, C-2), 8.11 (s, 1H, CH), 11.07 (s, 1H, NH); resolved signals for minor *antiperiplanar* conformer around the amide bond: 8.32 (s, 1H, CH), 11.00 (1H, s), ^13^C NMR (151 MHz, DMSO-d_6_): signals for major *synperiplanar* conformer around the amide bond: δ = 28.01 (CH_3_), 32.91 (NCH_3_), 41.38 (CH_2_), 55.11 (OCH_3_), 77.88 (OC), 103.66 (C-4), 109.95 (C-3), 111.05 (C-7), 112.10 (C-6), 125.01 (C-3a), 132.65 (C-7a), 134.03 (C-2), 140.20 (CH), 154.65 (C-5), 155.89 (OC=O), 169.72 (C=O). HRMS (ESI) *m*/*z*: calcd: [M+H]+ 361.187032. Found: [M+H]+ 361.1869.

***tert-butyl(2-{(2E)-2-[(1-methyl-1H-indol-3-yl)methylidene]hydrazinyl}-2-oxoethyl)carbamate,* 3l** Yield: 54%; m.p. 178–179 °C. ^1^H NMR (600 MHz, DMSO-d_6_): 1:0.40 mixture of conformers; signals for major *synperiplanar* conformer around the amide bond: δ = 1.41 (s, 9H, CH_3_), 3.82 (s, 3H, NCH_3_), 4.12 (d, J = 6.1 Hz, 2H, CH_2_), 6.83 (t, J = 6.1 Hz, 1H, NH), 7.22 (ddd, J = 0.7, 7.1, 7.8 Hz, 1H, H-5), 7.28 (ddd, J = 1.2, 7.0, 8.2 Hz, 1H, H-6), 7.51 (d, J = 7.5 Hz, 1H, H-7), 7.77 (s, 1H, C-2), 8.08 (d, J = 7.8 Hz, 1H, H-4), 8.13 (s, 1H, CH), 11.06 (s, 1H, NH); resolved signals for minor *antiperiplanar* conformer around the amide bond: 8.34 (s, 1H, CH), 11.02 (s, 1H, NH). ^13^C NMR (151 MHz, DMSO-d_6_): signals for major *synperiplanar* conformer around the amide bond: δ = 28.24 (CH_3_), 32.76 (NCH_3_), 41.42 (CH_2_), 77.87 (OC), 110.27 (C-7), 110.38 (C-3), 120.83 (C-5), 121.55 (C-4), 122.66 (C-6), 124.43 (C-3a), 133.90 (C-2), 137.56 (C-7a), 140.22 (CH), 155.96 (OC=O), 169.78 (C=O). HRMS (ESI) *m/z*: calcd: [M+H]+ 331.176467. Found: [M+H]+ 331.1764.

***tert-butyl(2-{(2E)-2-[(1-benzyl-1H-indol-3-yl)methylidene]hydrazinyl}-2-oxoethyl)carbamate,* 3m** Yield: 90%; m.p. 207–208 °C. ^1^H NMR (600 MHz, DMSO): 1:0.40 mixture of conformers; signals for major *antiperiplanar* conformer around the amide bond: δ 11.10 (s, 1H, NH), 8.15 (s, 1H, CH), 8.08 (d, *J* = 8.3 Hz, 1H, H-4), 7.95 (s, 1H, H-2), 7.50 (d, *J* = 7.4 Hz, 1H, H-7), 7.30 (d, *J* = 7.5 Hz, 2H, o-Ph), 7.26–7.22 (m, 1H, H-6), 7.21–7.18 (m, 1H, H-5), 7.24 (t, *J* = 8.7 Hz, 2H, m-Ph), 7.20 (t, *J* = 8.0 Hz, 1H, p-Ph), 6.85 (t, *J* = 6.1 Hz, 1H, NH), 5.44 (s, 2H, CH_2_), 4.10 (d, *J* = 6.1 Hz, 2H, CH_2_), 1.40 (s, 9H, CH_3_); resolved signals for minor *antiperiplanar* conformer around the amide bond: 7.05 (t, *J* = 6.0 Hz, 1H, NH), 8.21 (d, *J* = 7.9 Hz, 1H, H-4), 8.34 (s, 1H, CH), 3.62 (d, *J* = 6.1 Hz, 2H, CH_2_). ^13^C NMR (151 MHz, DMSO): signals for major *antiperiplanar* conformer around the amide bond: δ 170.03 (C=O), 156.16 (O-C=O), 140.39 (CH), 137.68 (i-Ph), 137.02 (C-7a), 133.54 (C-2), 128.79 (C-o), 127.72 (p-Ph), 127.27 (m-Ph), 124.85 (C-3a), 123.03 (C-5), 121.87 (C-4), 121.18 (C-6), 111.15 (C-3), 110.93 (C-7), 78.12 (C), 49.45 (CH_2_), 42.56 (CH_2_), 28.38 (CH_3_); resolved signals for minor *synperiplanar* conformer around the amide bond: 143.22 (CH), 122.29 (C-2), 78.27 (C), 41.57 (CH_2_), 28.35 (CH_3_). HRMS (ESI) *m*/*z*: calcd: [M+H]+ 407.207767. Found: [M+H]+ 407.2078.

***2-(1H-indol-3-yl)-N’-[(E)-(5-methoxy-1H-indol-3-yl)methylidene]acetohydrazide,* 3n**^1^H NMR, ^13^C NMR and HRMS spectra of compound **3n** were published elsewhere [65].

***N’-[(E)-(3,4-dimethoxyphenyl)methylidene]-1H-indole-3-carbohydrazide, 3o*** Yield: 78%; m.p. 267–270 °C. ^1^H NMR (400 MHz, DMSO-d_6_, 353K): δ = 3.83 (s, 3H, OCH_3_), 3.84 (s, 3H, OCH_3_), 7.03 (d, J = 8.3 Hz, 1H, H-5′), 7.14 (ddd, J = 1.3, 7.0, 7.7 Hz, 1H, H-5), 7.18 (dt, J = 1.6, 6.6 Hz, 1H, H-6), 7.20 (dd, J = 1.6, 7.2 Hz, 1H, H-6′), 7.35 (d, J = 1.9 Hz, 1H, H-2′), 7.48 (d, J = 8.0 Hz, 1H, H-7), 8.21 (d, J = 7.6 Hz, 1H, H-4), 8.26 (s, 1H, CH), 10.98 (bs, 1H, NH), 11.52 (bs, 1H, NH). ^13^C NMR (151 MHz, DMSO-d_6_, 353K): δ = 55.48 (CH_3_), 55.56 (CH_3_), 108.45 (C-3), 109.14 (C-2′), 111.45 (C-5′), 112.05 (C-7), 120.22 (C-4), 120.74 (C-5), 120.89 (C-6′), 121.73 (C-6), 126.53 (C-1′), 127.66 (C-3a), 129.17 (C-2), 135.69 (C-7a), 144.03 (CH), 149.15 (C-3′), 150.35 (C-4′), 162.13 (C=O). HRMS (ESI) *m*/*z*: calcd: [M+H]+ 324.134268. Found: [M+H]+ 324.13358.

***N’-[(E)-(3,4-dimethoxyphenyl)methylidene]-2-(1H-indol-3-yl)acetohydrazide,* 3p** Yield: 92%; m.p. 177–180 °C. ^1^H NMR (600 MHz, DMSO-d_6_): 1:0.80 mixture of conformers; signals for major *antiperiplanar* conformer around the amide bond: δ = 3.80 (s, 3H, OCH_3_), 3.81 (s, 3H, OCH_3_), 4.05 (s, 2H, CH_2_), 6.95 (t, J = 7.5 Hz, 1H, H-5), 7.01 (d, 8.5 Hz, 2H, H-5′), 7.06 (t, J = 7.0 Hz, 1H, H-6), 7.17 (dt, J = 1.8, 8.7 Hz, 2H, H-6′), 7.28 (d, J = 1.7 Hz, 1H, H-2′), 7.35 (d, J = 2.42 Hz, 1H, H-2), 7.36 (d, J = 9.27 Hz, 1H, H-7), 7.60 (dd, J = 3.46, 7.85 Hz, 1H, H-4), 7.91 (s, 1H, CH), 10.87 (bs, 1H, NH), 11.18 (s, 1H, NH); resolved signals for minor *antiperiplanar* conformer around the amide bond: 8.15 (s, 1H, CH), 10.91 (bs, 1H, NH), 11.40 (s, 1H, NH) ^13^C NMR (151 MHz, DMSO-d_6_): signals for major *antiperiplanar* conformer around the amide bond: δ = 29.24 (CH_2_), 55.40 (OCH_3_), 108.22 (C-2′), 108.25 (C-3), 111.30 (C-5′), 111.56 (C-7), 118.28 (C-4), 118.66 (C-5), 120.97 (C-6′), 121.59 (C-6), 123.86 (C-2), 127.13 (C-1′), 127.39 (C-3a), 135.98 (C-7a), 142.46 (CH), 148.99 (C-3′), 150.56 (C-4′), 172.52 (C=O). HRMS (ESI) *m*/*z*: calcd: [M+H]+ 338.149918. Found: [M+H]+ 338.14902.

***N’-[(E)-(4-hydroxy-3-methoxyphenyl)methylidene]-2-(1H-indol-3-yl)acetohydrazide,* 3q** Yield: 72%; m.p. 246–247 °C. ^1^H-NMR (400 MHz, DMSO-d_6_): 0.54: 0.46 mixture of two conformers; signals for major conformer: δ = 3.81 (s, 3H, CH_3_), 4.04 (s, 2H, CH_2_), 6.82 (d, J = 8.1 Hz, 1H, H-5′), 6.96–7.00 (m, 1H, H-5), 7.02–7.09 (m, 2H, H-6, and H-6‘), 7.22 (d, J = 2.3 Hz, 1H, H-2), 7.29 (d, J = 1.8 Hz, 1H, H-2′), 7.33 (d, J = 6.8 Hz, 1H, H-7), 7.59 (d, J = 7.9 Hz, 1H, H-4), 7.87 (s, 1H, CH), 9.46 (s, 1H, OH), 10.86 (s, 1H, NH), 11.10 (s, 1H, NH); Signals for minor conformer: δ(ppm): 3.61 (s, 2H, CH_2_), 3.79 (s, 3H, CH_3_), 6.81 (d, J = 8.1 Hz, 1H, H-5′), 6.93–6.97 (m, 1H, H-5), 7.02–7.09 (m, 2H, H-6, and H-6‘),7.24 (d, J = 1.9 Hz, 2H, H-2, and H-2′), 7.35 (d, J = 7.0 Hz, 1H, H-7), 7.59 (d, J = 7.9 Hz, 1H, H-4), 8.11 (s, 1H, CH), 9.48 (s, 1H, OH), 10.90 (s, 1H, NH), 11.33 (s, 1H, NH). ^13^C-NMR (151 MHz, DMSO-d_6_): 0.54:0.46 mixture of two conformers; signals for major conformer: δ = 29.20 (CH_2_), 55.51 (CH_3_), 108.33 (C-3), 109.10 (C-2′), 111.30 (C-7), 115.37 (C-5′), 118.27 (C-5), 118.68 (C-4), 120.88 (C-6′), 121.11 (C-6), 123.86 (C-2), 125.82 (C-1′), 127.42 (C-4a), 135.98 (C-7a), 142.81 (CH), 147.97 (C-3′), 148.51 (C-4′), 172.43 (C=O); signals for minor conformer: δ(ppm): 31.69 (CH_2_), 55.51 (CH_3_), 108.31 (C-3), 108.93 (C-2′), 111.35 (C-7), 115.52 (C5′), 118.36 (C-5), 118.68 (C-4), 121.00 (C-6′), 121.87 (C-6), 123.86 (C-2), 125.73 (C-1′), 127.14 (C-4a), 136.10 (C-7a), 146.65 (CH), 147.97 (C-3′), 148.78 (C-4′), 166.79 (C=O). HRMS (ESI) *m*/*z*: calcd: [M+H]+ 324.134268. Found: [M+H]+ 324.1341.

***2,4-dihydroxy-N’-[(E)-(4-hydroxy-3-methoxyphenyl)methylidene]benzohydrazide,* 3r** Yield: 76%; m.p. 242–243 °C. ^1^H NMR (400 MHz, DMSO-d_6_): 3.36 (s, 3H, CH_3_), 6.31 (d, J = 2.4 Hz, 1H, H-3), 6.37 (dd, J = 2.4, 8.7 Hz, 1H, H-5), 6.85 (d, J = 8.1 Hz, 1H, H-6′), 7.10 (dd, J = 1.8, 8.1 Hz, 1H, H-5′), 7.32 (d, J = 1.8 Hz, 1H, H-2′), 7.80 (d, J = 8.7 Hz, 1H, H-6), 8.33 (s, 1H, CH), 9.56 (s, 1H, OH), 10.19 (s, 1H, OH), 11.54 (s, 1H, NH), 12.45 (s, 1H, OH). ^13^C NMR (100 MHz, DMSO-d_6_): 55.55 (CH_3_), 102.86 (C-3), 106.20 (C-1), 107.33 (C-5), 108.99 (C-2′), 115.45 (C-6′), 122.27 (C-5′), 125.59 (C-1′), 129.49 (C-6), 148.05 (C-4′), 148.61 (CH), 149.09 (C-3′), 162.31 (C-4), 162.57 (C-2), 165.28 (C=O). HRMS (ESI) *m/z*: calcd: [M+H]+ 303.097548. Found: [M+H]+ 303.0974.

### 3.3. General Procedure for the Synthesis of Compounds ***5a–5l***

The solution of 20 mmol of the corresponding carbonyl compounds (**1a**, **1i**, **1j**, **1k**, **1l**, **1o**) in 10 mL of absolute ethanol was mixed with a hot solution of 20 mmol (60 °C) benzenesulfonohydrazide (**4a**) or 4-methylbenzenesulfonohydrazide (**4b**) and 4-methoxylbenzenesulfonohydrazide (**4c**) in 10 mL of absolute ethanol and stirred for 1–3 h. The obtained crystalline precipitates were filtered, washed with ethanol-ether, recrystallized from ethanol.

***N’-[(E)-(1-benzylpiperidin-4-yl)methylidene]benzenesulfonohydrazide,* 5a** Yield: 59%; m.p. 110–111 °C. ^1^H NMR (400 MHz, DMSO-d_6_): δ = 1.26–1.36 (m, 2H, CH_2_), 1.54–1.58 (m, 2H, CH_2_), 1.91–1.97 (m, 2H, CH_2_), 2.05–2.14 (m, 1H CH), 2.65–2.68 (m, 2H, CH_2_), 3.40 (s, 2H, CH_2_), 7.19 (d, J = 4.8 Hz, 1H, CH), 7.21–7.27 (m, 3H, H-2′’, H-4′’ and H-6′’), 7.29–7.33 (m, 2H, H-3′’, and H-5′’), 7.59–7.63 (m, 2H, H-3′, and H-5′), 7.65–7.69 (m, 1H, H-4′), 7.78–7.81 (m, 2H, H-2′, and H-6′), 10.93 (1H, s).^13^C NMR (DMSO) (100 MHz, DMSO-d_6_): δ = 27.62 (CH_2_), 36.94 (CH), 51.11 (CH_2_), 61.25 (CH_2_), 125.85 (C-4″), 126.12 (C-2′ and C-6′), 127.11 (C-3″ and C-5″), 127.75 (C-2′’ and C-6′’), 128.07 (C-3′ and C-5′), 131.86 (C-4′), 137.28 (C-1′’), 137.94 (C-1′), 153.47 (CH).HRMS (ESI) *m/z*: calcd: [M+H]+ 358.158373. Found: [M+H]+ 358.1581.

***4-methoxy-N’-[(E)-(5-methoxy-1-methyl-1H-indol-3-yl)methylidene]benzenesulfonohydrazide,* 5b** Yield: 78%; m.p. 145–146 °C. ^1^H NMR (600 MHz, DMSO-d_6_): δ = 3.74 (s, 3H, NCH_3_), 3.77 (s, 3H, OCH_3_), 3.80 (s, 3H, OCH_3_), 6.86 (dd, J = 2.6, 8.8 Hz, 1H, H-6), 7.11 (d, J = 9.0 Hz, 2H, H-3′, and H-5′), 7.36 (d, J = 8.8 Hz, 1H, H-7), 7.47 (d, J = 2.5 Hz, 1H, H-4), 7.64 (s, 1H, H-2), 7.85 (d, J = 8.8 Hz, 2H, H-2′, and H-6′), 8.05 (s, 1H, CH), 10.75 (s, 1H, NH). ^13^C NMR (151 MHz, DMSO-d_6_): δ = 32.90 (NCH_3_), 55.22 (OCH_3_), 55.64 (OCH_3_), 103.31 (C-4), 109.71 (C-3), 111.01 (C-7), 112.54 (C-6), 114.20 (C-3′ and C-5′), 124.90 (C-3a), 129.49 (C-1′), 130.83 (C-2′ and C-6′), 132.52 (C-7a), 134.20 (C-2), 145.01 (CH), 154.66 (C-4), 162.45 (C-4′). HRMS (ESI) *m/z*: calcd: [M+H]+ 374.11597. Found: [M+H]+ 374.1169.

***4-methoxy-N’-[(E)-(5-methoxy-1H-indol-3-yl)methylidene]benzenesulfonohydrazide,* 5c** Yield: 53%; m.p. 159–163 °C. ^1^H NMR (400 MHz, DMSO-d_6_): δ = 3.76 (s, 3H, OCH_3_), 3.80 (s, 3H, OCH_3_), 6.80 (dd, J = 2.6, 8.8 Hz, 1H, H-6), 7.11 (d, J = 9.0 Hz, 2H, H-3′, and H-5′), 7.29 (d, J = 8.8 Hz, 1), 7.46 (d, J = 2.6 Hz, 1H, H-4), 7.66 (d, J = 2.8 Hz, 1H, H-2), 7.85 (d, J = 9.0 Hz, 2H, H-2′ and H-6′), 8.08 (s, 1H, CH), 10.76 (s, 1H, NH), 11.37 (d, J = 2.2 Hz, 1H, NH).^13^C NMR (100 MHz, DMSO-d_6_): δ = 55.15 (OCH_3_), 55.63 (OCH_3_), 103.12 (C-4), 110.85 (C-3), 112.48 (C-7), 112.59 (C-6), 114.19 (C-3′ and C-5′), 124.49 (C-3a), 129.48 (C-1′), 130.73 (C-2′ and C-6′), 130.84 (C-7a), 131.78 (C-2), 145.46 (CH), 154.35 (C-4), 162.43 (C-4′). HRMS (ESI) *m/z*: calcd: [M+H]+ 360.10125. Found: [M+H]+ 360.10037.

***N’-[(E)-(1-benzyl-1H-indol-3-yl)methylidene]benzenesulfonohydrazide,* 5d** Yield: 89%; m.p. 146–149 °C. ^1^H NMR (600 MHz, DMSO-d_6_): δ = 5.41 (s, 3H, CH_2_), 7.15 (t, J = 7.4 Hz, 1H, H-5), 7.18 (ddd, J = 1.2, 7.1, 8.2 Hz, 1H, H-6), 7.21 (d, J = 7.0 Hz, 2H, H-2′, and H-6′), 7.24 (t, J = 7.3 Hz, 1H, H-4′), 7.30 (t, J = 7.3 Hz, 2H, H-3′, and H-5′), 7.48 (d, J = 8.1 Hz, 1H, H-7), 7.59–7.65 (m, 3H, H-3″, H-4″, and H-5″), 7.90 (s, 1H, H-2), 7.93 (dd, J = 1.0, 7.6 Hz, 2H, H-2″, and H-6″), 7.97 (d, J = 7.8 Hz, 1H, H-4), 8.10 (s, 1H, CH), 11.01 (s, 1H, NH).^13^C NMR (151 MHz, DMSO-d_6_): δ = 49.27 (CH_2_), 110.65 (C-3), 110.67 (1C, s), 120.89 (C-5), 121.78 (C-4), 122.79 (C-6), 124.61 (C-3a), 127.09 (C-2′ and C-6″), 127.28 (C-2″ and C-6″), 127.51 (C-4′), 128.59 (C-3′ and C-5′), 129.05 (C-3″ and V-5″), 132.82 (C-4″), 133.50 (C-2), 136.75 (C-7a), 137.44 (C-1′), 139.13 (C-1″), 144.60 (CH). HRMS (ESI) *m/z*: calcd: [M+H]+ 390.127073. Found: [M+H]+ 390.12604.

***4-methoxy-N’-[(E)-(1-methyl-1H-indol-3-yl)methylidene]benzenesulfonohydrazide,* 5e** Yield: 66%; m.p. 170–175 °C. ^1^H NMR (600 MHz, DMSO-d_6_): δ = 3.78 (s, 3H, NCH_3_), 3.79 (s, 3H, OCH_3_), 7.11 (d, J = 9.0 Hz, 2H, H-3′ and H-5′), 7.17 (t, J = 7.5 Hz, 1H, H-5), 7.24 (ddd, J = 1.1, 7.1, 8.1 Hz, 1H, H-6), 7.46 (d, J = 8.2 Hz, 1H, H-7), 7.69 (s, 1H, H-2), 7.84 (d, J = 9.0 Hz, 2H, H2′, and H-6′), 7.97 (d, J = 7.9 Hz, 1H, H-4), 8.05 (s, 1H, CH), 10.76 (s, 1H, NH). ^13^C NMR (151 MHz, DMSO-d_6_): δ = 32.73 (NCH_3_), 55.61 (OCH_3_), 110.11 (C-3), 110.17 (C-7), 114.18 (C-3′ and H-5′), 120.74 (C-5), 121.66 (C-4), 122.62 (C-6), 124.40 (C-3a), 129.48 (C-2′ and C-6′), 130.81 (C-2′ and C-6′), 133.89 (C-1′), 137.42 (C-7a), 144.44 (CH), 162.44 (C-4′).HRMS (ESI) *m/z*: calcd: [M+H]+ 344.106338. Found: [M+H]+ 344.1055.

***N’-[(E)-(5-methoxy-1-methyl-1H-indol-3-yl)methylidene]benzenesulfonohydrazide,* 5f**^1^H NMR, ^13^C NMR and HRMS spectra of compound **5f** were published elsewhere [66].

***N’-[(E)-(1-benzyl-1H-indol-3-yl)methylidene]-4-methylbenzenesulfonohydrazide,* 5g** Yield: 79%; m.p. 145–146 °C.^1^H NMR (400 MHz, DMSO-d_6_): δ = 2.34 (s, 3H, CH_3_), 5.41 (s, 2H, CH_2_), 7.13–7.32 (m, 7H, H-5, H6, H-2″, H-3″, H-4″, H-5″, and H-6″), 7.39 (d, J = 7.9 Hz, 2H, H-3′, and H-5′), 7.48 (dd, J = 1.3, 7.1 Hz, 1H, H-7), 7.81 (d, J = 8.2 Hz, 2H, H-2′, and H-6′), 7.89 (s, 1H, H-2), 7.99 (d, J = 7.3 Hz, 1H, H-4), 8.08 (s, 1H, CH), 10.92 (s, 1H, NH).^13^C NMR (100 MHz, DMSO-d_6_): δ = 20.95 (CH_3_), 49.26 (CH_2_), 110.66 (C-7), 110.72 (C-3), 120.88 (C-5), 121.83 (C-4), 122.78 (C-6), 124.63 (C-4a), 127.08 (C-2″ and C-6″), 127.33 (C-2′ and C-6′), 127.51 (C-4″), 128.59 (C-3″ and C-5″), 129.47 (C-3′ and C-5′), 133.41 (C-2), 136.26 (C-1′), 136.74 (C-7a), 137.46 (C-1″), 143.16 (C-4′), 144.32 (C-H). HRMS (ESI) *m/z*: calcd: [M+H]+ 404.142723. Found: [M+H]+ 404.1425.

***N’-[(E)-(3,4-dimethoxyphenyl)methylidene]-4-methylbenzenesulfonohydrazide,* 5h**^1^H NMR, ^13^C NMR and HRMS spectra of compound **5h** were published elsewhere [66].

***N’-[(E)-(*3*,*4*-dimethoxyphenyl)methylidene]benzenesulfonohydrazide,* 5i**^1^H NMR, ^13^C NMR and HRMS spectra of compound **5i** were published elsewhere [66].

***N’-{(E)-[5-(benzyloxy)-1H-indol-3-yl]methylidene}-4-methylbenzenesulfonohydrazide,* 5j** Yield: 67%; m.p. 222–223 °C. ^1^H NMR (400 MHz, DMSO-d_6_): δ = 2.28 (s, 3H, CH_3_), 5.03 (s, 2H, CH_2_), 6.88 (dd, J = 2.5, 8.8 Hz, 1H, H-6), 7.30 (d, J = 8.8 Hz, 1H, H-7), 7.34 (d, J = 8.1 Hz, 2H, H-3′, and H-5′), 7.36 (t, J = 7.2 Hz, 1H, H-4″), 7.43 (t, J = 7.4 Hz, 2H, H-3″, and H-5″), 7.53 (d, J = 7.2 Hz, 2H, H-2″, and H-6″), 7.57 (d, J = 2.4 Hz, 1H, H-4), 7.68 (d, J = 2.8 Hz, 1H, H-2), 7.81 (d, J = 8.2 Hz, 2H, H-2′, and H-6′), 8.09 (s, 1H, CH), 10.84 (s, 1H, NH), 11.40 (d, J = 2.0 Hz, 1H, NH). ^13^C NMR (100 MHz, DMSO-d_6_): δ = 20.90 (CH_3_), 69.68 (CH_2_), 104.82 (C-4), 110.81 (C-3), 112.50 (C-6), 113.02 (C-7), 124.48 (C-4a), 127.37 (C-2′ and C-6′), 127.84 (C-4″), 127.93 (C-2″ and C-6″), 128.46 (H-3″ and H-5″), 129.41 (C-3′ and C-5′), 130.96 (C-2), 131.98 (C-7a), 136.33 (C-1′), 137.35 (C-1″), 143.08 (C-4′), 145.68 (CH), 153.38 (C-5). HRMS (ESI) *m/z*: calcd: [M+H]+ 404.142723. Found: [M+H]+ 420.1373.

***N’-((E)-[5-(benzyloxy)-1H-indol-3-yl]methylidenebenzenesulfonohydrazide*, 5k**^1^H NMR, ^13^C NMR and HRMS spectra of compound **5k** were published elsewhere [67].

***N’-[(E)-(4-chlorophenyl)methylidene]benzenesulfonohydrazide,* 5l**^1^H NMR, ^13^C NMR and HRMS spectra of compound **5l** were published elsewhere [66].

### 3.4. Biological Evaluation

#### Assessment of AChE and BChE Inhibitory Activity

AChE and BChE inhibitory activity was measured using the microplate assay described by Ellman et al. [68] with the modifications added by López et al. [36]. The compounds were tested at concentrations between 10^−3^ and 10^−8^ M. First, all compounds were dissolved in 1% DMSO in a concentration of 1 mg/mL. Then, they were serially diluted using phosphate buffer (PBS) (8 mM K_2_HPO_4_, 2.3 mM NaH_2_PO_4_, 0.15 M NaCl, pH 7.5) to provide the concentration range needed. Acetylcholinesterase from *Electrophorus electricus* and butyrylcholinesterase from equine serum (Sigma-Aldrich, Hamburg, Germany) were used with a substrate solution of 5,5′-dithiobis(2-nitrobenzoic acid) (DTNB) with acetylthiocholine iodide (ATCI) or butyrylthiocholine iodide (BuTCI), respectively (0.04 M Na_2_HPO_4_, 0.2 mM DTNB, 0.24 mM ATCI or BuTCI, pH 7.5).

Then, 50 microliters of AchE or BchE (0.25 U/mL) dissolved in phosphate buffer and 50 μL of the tested compound solution were added to the wells. Incubation of the plates was performed at room temperature for 30 min. Then, 100 μL of substrate solution were added to start the enzymatic reaction. The absorbances were read in a microplate reader (BIOBASE, ELISA-EL10A, China) at 405 nm after 3 min for AChE and 10 min for BChE. Enzyme activity was calculated as an inhibition percentage compared to an assay including buffer instead of an inhibitor. Galanthamine and donepezil were used as positive controls. All data were analyzed with the software package Prism 3 (Graph Pad Inc., San Diego, CA, USA). The IC_50_ values were measured in triplicate and the results are presented as means ± SD. For the less active compounds instead of IC_50_, an inhibition percentage of 1 mM is presented (1 mM is the maximal tested concentration).

### 3.5. Cytotoxicity of the Compounds

#### 3.5.1. Cell Lines and Culture Conditions

To evaluate the biocompatibility of the experimental compounds, their in vitro cytotoxicity was assessed against malignant neuroblastoma cells of human (SH-SY5Y) and murine (NEURO-2A) origin, as well as normal fibroblast murine cells (CCL-1). All cell lines were purchased from the German Collection of Microorganisms and Cell Cultures (DSMZ GmbH, Braunschweig, Germany). Cell cultures were cultivated in a growth medium RPMI 1640 supplemented with 10% fetal bovine serum (FBS), 5% L-glutamine, and incubated under standard conditions of 37 °C and 5% humidified CO2 atmosphere.

#### 3.5.2. MTT Assay

The effects of the newly synthesized hydrazide hydrazone and sufonyl hydrazone derivatives on cell viability were measured using a standard MTT colorimetric assay. The method is based on the biotransformation of the yellow tetrazole salt MTT (3-(4,5-dimethyliazol-2-yl)-2,5-diphenyltetrazole bromide) to violet formazan under the action of mitochondrial succinate dehydrogenases of vital cells. Exponential phased cells were harvested and seeded (100 μL/well) in 96-well plates at the appropriate density (1.5 × 10^5^). On the following day, cell cultures were treated with different concentrations of the experimental compounds and after 72 h exposure time, filter sterilized MTT substrate solution (5 mg/mL in PBS) was added to each well of the culture plate. A further 4 h incubation allowed for the formation of purple insoluble formazan precipitates. The latter were dissolved in isopropyl alcohol solution containing 5% formic acid prior to absorbance measurement at 580 nm on a Labexim LMR1 automated ELISA reader. Collected absorbance values were blanked against MTT and isopropanol solution and normalized to the mean value of untreated control (100% cell viability).

### 3.6. Determination of Antioxidant Activity

#### 3.6.1. DPPH Radical Scavenging Activity

Free radical scavenging activity was measured using DPPH method [69,70] with slight modification. Here, 300 μL of 5 mM solutions of tested compounds in MeOH were added to 2 mL methanolic solution of DPPH (2 mg/mL). The absorbance was measured at 517 nm after 30 min. Results were evaluated as percentage scavenging of radical:DPPH radical scavenging activity (%) = ((Abs_contr_. − Abssample)/Ab_scontr_.) ×100,
where Ab_scontr_. is the absorbance of DPPH radical with 300 μL MeOH, Abssample is the absorbance of DPPH radical solution mixed with sample. BHT was used as positive control.

#### 3.6.2. ABTS Radical Scavenging Assay

For ABTS assay, the procedure followed the method of Arnao et al. [71] with some modifications. The stock solutions included 7 mM ABTS solution and 2.4 mM potassium persulfate solution. The working solution was then prepared by mixing the two stock solutions in equal quantities and allowing them to react for 14 h at room temperature in the dark. The solution was then diluted by mixing 2 mL ABTS solution with 50 mL methanol to obtain an absorbance of 0.605 ± 0.01 units at 734 nm using a spectrophotometer. A fresh ABTS solution was prepared for each assay. Different concentrations (5, 2.5, 1, 1.25, 0.5 mM) (10 μL) of compounds were allowed to react with 2 mL of the ABTS solution and the absorbance was taken at 734 nm after 5 min. The ABTS scavenging capacity of the compound was calculated as follows:ABTS radical scavenging activity (%) = ((Abs_contr_. − Abs_sample_)/Abs_contr_.) ×100,
where Abs_contr_. is the absorbance of DPPH radical with 300 μL MeOH, Abssample is the absorbance of DPPH radical solution mixed with the sample. The IC_50_ value (concentration of sample where absorbance of ABTS decreases 50% with respect to absorbance of blank) of the sample was determined. BHT was used as positive control.

#### 3.6.3. Ferric Reducing/Antioxidant Power (FRAP)

The FRAP assay was done according to the method described by Benzie and Strain [72] with some modifications. The stock solutions included 300 mM acetate buffer pH 3.6, 10 mM TPTZ solution in 40 mM HCl, and 20 mM FeCl_3_.6H_2_O solution. The fresh working solution was prepared by mixing 25 mL acetate buffer, 2.5 mL TPTZ solution, and 2.5 mL FeCl_3_.6H_2_O solution and then warmed at 37 °C before use. Then, 150 μL of compounds in MeOH (5 mM) were allowed to react with 2 mL of the FRAP solution for 30 min in the dark. Readings of the colored product (ferrous tripyridyltriazine complex) were then taken at 593 nm. Results are expressed in mM Trolox equivalent (TE/mM). BHT was used as positive control.

#### 3.6.4. Determination of Antioxidant Activity in Linoleic Acid System by the FTC Method

The antioxidant activity of studied compounds against lipid peroxidation was measured through ammonium thiocyanate assay as described by Takao et al. [66] with some modifications. The reaction solution, containing 200 μL of the compound (10 mM) in MeOH, 200 μL of linoleic acid emulsions (25 mg/mL in 99% ethanol), and 400 μL of 50 mM phosphate buffer (pH 7.6) was incubated in the dark at room temperature. A 10 μL aliquot of the reaction solution was then added to 200 μL of 70% (*v*/*v*) ethanol and 10 μL of 30% (*w*/*v*) ammonium thiocyanate. Precisely 3 min after the addition of 10 μL of 20 mM ferrous chloride in 3.5% (*v*/*v*) hydrochloric acid to the reaction mixture, the absorbance of the resulting red color was measured at 500 nm. Aliquots were assayed every 24 h until the day after the absorbance of the control solution (without compound) reached maximum value. BHT (10 mM) was used as positive control.

### 3.7. Model of H_2_O_2_-Induced Oxidative Stress in SH-SY5Y Cell Line

The human neuroblastoma cell line SH-SY5Y (94030304) was acquired from the European Collection of Cell Cultures (ECACC, Salisbury, UK). SH-SY5Y cells were cultured in RPMI medium supplemented with 10% heat-inactivated fetal bovine serum (FBS), 2 mM L-glutamine, and penicillin/streptomycin. The cells were maintained in a humidified atmosphere at 37 °C with 5% CO2. The culture medium was refreshed every 2–3 days to ensure optimal conditions for cell growth and viability.

#### 3.7.1. Model

The SH-SY5Y cells were seeded at a density of 3.5×104 cells per well (100 μL of RPMI) in 96-well plates. After 24 h, the cell medium was aspirated and the cells were treated with different concentrations of the test compounds (0.1, 1, 5, 10, 25, 50 μM) for 90 min. Thereafter, SH-SY5Y cells were washed with phosphate-buffered saline (PBS) and exposed to hydrogen peroxide (1 mM) for 10 min. The solution of H_2_O_2_ in PBS was aspirated and changed with cell medium. After 24 h, cell viability was evaluated by the MTT assay. Cells not treated with hydrogen peroxide (negative controls) were considered a measure for 100% protection, and cells treated with hydrogen peroxide (positive controls) for 0% protection. Melatonin and rasagiline (Sigma-Aldrich Chemie GmbH) were used as references due to the abundance of data on their antioxidant and neuroprotective activity [73,74]. Considering the structural similarity of the newly synthesized compound to donepezil, it was included as a reference compound in the study.

#### 3.7.2. Statistical Analysis

Statistical analysis was conducted using GraphPad Prism 6 Software. A one-way ANOVA followed by Dunnett’s multiple comparisons post-test was utilized to compare the data between the control and treatment groups. A significance level of 0.05 was selected for all comparisons.

### 3.8. In Silico Studies

#### 3.8.1. Molecular Docking of Human AChE and of Human BChE

Molecular docking studies were performed using Docking tool of Molecular Operating Environment of Chemical Computing Group (MOE, version 2022.02).

Selection of PDB structures: The following crystallographic structures of human acetylcholinesterase (AChE) and human butyrylcholinesterase (BChE) have been chosen for molecular docking studies:(1)AChE, complexed with 1-Benzyl-4-[(5,6-dimethoxy-1-indanon-2-yl)methyl]piperidine (E20) and co-factor 2-acetamido-2-deoxy-beta-D-glucopyranose (NAG), retrieved from Protein Data Bank (http://www.rcsb.org/) with PDB ID 4EY7;(2)BChE, complexed with butyl-[(2~(S))-1-(2-cycloheptylethylamino)-3-(1~(H)-indol-3-yl)-1-oxidanylidene-propan-2-yl]azanium (HUN) and NAD again (PDB ID 6QAA).

The choice of aforementioned crystallographic structures followed previous investigations of Alov et al. [46], who, aiming at multi-target hit compounds for neurodegenerative disease drug development, performed molecular docking in AChE (PDB ID 4EY7). Aktar et al. [75] have investigated the enzyme inhibition activities of new sulfonyl hydrazones performing molecular docking simulations in BChE (PDB ID 6QAA).

Structures preparation: The 3D structures of the investigated compounds were built and optimized with AMBER10:EHT in MOE. The R enantiomers (a chiral carbon atom in the pyrrolidine ring) of compounds **3e–3i** were used in docking. These forms correspond to the extended conformations of the compounds, thus fitting better to the deep and narrow binding cavity of the cholinesterases. The PDB complexes were prepared using the “Quick preparation” procedure in MOE. In addition to adding hydrogens and assigning protonation states of titratable protein groups by the Generalized Born electrostatics model at physiologically relevant conditions (temperature of 300 K; pH = 7; ion concentration of 0.1 mol/L), tethers were added to the receptor heavy atoms and distant atoms were fixed. Following this, the system refinement was performed applying Amber10: EHT force field and the default RMS gradient of 0.1 kcal/mol/Å^2^.

Prediction of the protonation state of the investigated structures. The “Protomer” panel in MOE was used to explore the possible ionization states of the molecules that can be populated at physiological pH. The resulting ionization states were sorted in order of decreasing population and the ionization state with the highest % of population at the physiological pH was used for docking.

Docking simulations: Molecular docking studies were performed in the “Docking” module in MOE. The following settings were applied in the docking protocol: (I) in the set of atoms defining the receptor, the receptor and solvent atoms were included. (II) “Triangle Matcher” placement to generate poses by aligning ligand triplets of atoms on triplets of alpha spheres in the most systematic way; the London dG scoring function was used to rank the poses (30 placement poses). (III) Subsequent refinement using “Rigid receptor” or “Induced fit” (flexible side chains) tools based on London dG and GBVI/WSA scoring functions (5 poses as the final output following the refinement step). London dG is the default scoring function in MOE that estimates the free energy of binding of the ligand from a given pose combining enthalpy and entropy terms. The GBVI/WSA dG is the default force field-based scoring function that is used after rescoring for the final refinement of the docking poses. The protein–ligand interactions in the active site of the complexes were visualized using the “Ligand Interactions” MOE tool.

#### 3.8.2. In Silico Docking Studies of MT1 and MT2 Receptors

We prepared a model of human melatonin receptors MT1 and MT2 which we employed as docking templates in order to propose the possible molecular mechanism of action of our ligands. XRD structures of human MT1 and MT2 receptors were selected for our modeling on the base of resolution and completeness of crystallized deposited data in Protein Data Bank.

If more than one copy of the receptor structure in the PDB existed, we used all of them in further analysis.

In all selected PDB structures, the minor structural improprieties were corrected, followed by removal of all nonprotein species. For the purpose of attaining the right protonation state at 7.0 pH, which was the physiological one, we used Labute’s protonate 3D algorithm (Labute 2008) at 300K and 0.154 M/l salt concentration as it is implemented in MOE software package.

The codes of selected PDB for MT1 were 6ME2, 6ME3, 6ME4, 6ME5, 7DB6, 7VGY, 7VGZ.

Although only 2 amino acids in the active site differ between these structures ALA104 in 6ME2, 6ME3, 6ME4, 6ME5 is GLY104 in 7DB6, 7VGY, 7VGZ and PHE251 in 6ME2, 6ME3, 6ME4, 6ME5 is TRP251 in 7DB6, 7VGY, 7VGZ, we prepared our model for docking by homology modeling the sequence of NCBI NP005949.1 melatonin receptor type 1A [Homo Sapiens] on consensus model formed by all of the PDB structures. The above-mentioned differences are due to the point mutations for increased thermostability of the protein introduced by Stauch et al. [51].

Active site for docking was defined by natural position of ligands inside crystallized structures.

The PDB codes of selected structures for MT2 were 6ME6, 6ME7, 6ME8, 6ME9, 6PS8, 7VH0. In this set, only PHE264 from the residues forming active site is replaced with TRP264 in 7VH0, based on the point mutations introduced by Stauch et al. [51].

In the case of MT2 receptor, we prepared our model for further docking by homology modeling the sequence of NCBI NP.005950.1 melatonin receptor 1B [Homo Sapiens] on consensus model formed by all of the selected PDB structures.

Fortunately, all of the selected XRD structures were crystallized with their natural-like ligands in the cavity, therefore presenting the active state of MT1 and MT2 receptors.

All newly synthesized ligands were protonated according to their protonation state at 7 pH. Due to the low amount of rotatable bonds, the systematic approach was used in formation of ligand library. Every generated structure was further energy minimized using AMBER12EHT [MOE] force field in gas phase. All unique conformational structures within 10 kcal/mol from the lowest structure of every ligand were used for further docking.

All conformations of all ligands were docked in active site of the MT1 and MT2 receptors using AlphaPMI algorithm [MOE] for initial placement of every structure. These poses were scored by London dG [MOE] function, which estimates the free energy of binding of the ligand from a given pose and consists of terms that estimate average gain/loss of rotational and translational entropy, measures the geometric imperfections of hydrogen bonds and the desolvation energy of atoms.

The best 100 poses for every ligand for every pocket were further optimized with Induced Fit methodology using AMBER12EHT force field and Generalized Born solvation model with optimization cutoff of 5A from the ligand. The GBVI/WSA dG [MOE] was used as a rescoring function and the best 30 poses were collected for the next coming analysis.

In all cases, the original ligands from the PDB files were also used in our docking study as they were subjected to the same ligand preparation methodology as our ligands. Placement of the original ligands in the receptor cavities from the placement algorithm in a position similar to that in the intact PDB was used as a criterion for the adequacy of our docking methodology.

RMSD of alpha carbons of the active site residues between aligned chains of selected PDB structures for MT1 and MT2 structures are presented in Appendix A. Differences in sequences of selected structures according to BLOSSUM62 are presented also in Appendix A.

The difference between *Z* and *E* isomers in some of the ligands discussed in the text was calculated using B3LYP/6–311++G** hybrid functional and basis set as they are implemented in GAUSSIAN 09 software suite.

As a verification of our docking study, we included in our ligand set the original ligands from the used pdb files. These ligands were treated in the same way as our structures and were used to expand the ligand set for docking. Analysis of their interaction energies shows that all the structures studied have better interaction energies than the original ligands and the docking procedure used positions the original ligands in the receptor cavities in the same way as they are in the original PDB files.

#### 3.8.3. ADME/Tox

In silico ADME studies were performed by using the online tool SwissADME/Tox of the Swiss Institute of Bioinformatics (https://www.sib.swiss, accessed on 20 April 2022).

#### 3.8.4. In Silico Prediction of BBB Permeability

The webserver was used to predict BBB permeability of compounds, SwissADME (https://www.sib.swiss, accessed on 20 April 2022).

#### 3.8.5. BBB Permeability by In Vitro PAMPA Test

The blood–brain barrier permeabilities were measured by PAMPA Permeability Analyzer (pIONInc, Billerica, MA, USA). The tested compounds and reference standards were prepared as 10 mM stock solutions in DMSO and further diluted in Prisma HT buffer pH 7.4. The experiment followed the BBB Protocol: 200 μL sample’s aliquots in the donor compartment, Brain Sink Buffer aliquots (200 μL) in the acceptor wells, BBB-1 lipid for coating the permeation membranes, “sandwich” assembly with the acceptor plate on top of the donor one, and incubation at temperature 25 °C for 4 h under humidity control without stirring. For permeability assessment, UV spectra of the blank, reference, donor, and acceptor plates were collected at wavelength 250–500 nm. Effective permeability Pe (10^−6 ^cm/s) of the compounds was calculated and presented as −log*Pe.* Samples were analyzed in triplicate and average values were reported. Highly permeable compounds were indicated by values of −logPe < 5, the medium permeable by −logPe between 5 and 6, and if −logPe > 6, the compound was considered as low permeable [76]. Theophylline, corticosterone, and propranolol HCl were used as reference standards to control the quality and consistency of the BBB permeability experiment [77].

## 4. Conclusions

Alzheimer’s disease (AD) is a significant concern for chemistry researchers seeking new multi-target compounds to address this complex neurodegenerative condition. Focusing on melatonin/donepezil hybrids as potential candidates for AD treatment, we synthesized 30 new derivatives. Among them, compounds **3c** and **3d** displayed the most promising AChE inhibitory activity (10.76 ± 1.66 μM and 9.77 ± 0.76 μM, respectively), while compound **3n**, containing melatonin, exhibited notable BChE inhibitory activity (21.12 ± 1.48 μM).

The in vitro antioxidant activities for the most promising molecules were investigated by the DPPH, FRAP, ABTS, and FTC methods. All tested compounds hold potential as therapeutic neuroprotective agents for neurodegenerative disorders. The sulfonylhydrazones **5k**, **5j**, **5g**, and hydrazone 3r revealed the highest DPPH activity and hydrazones **3i**, **3d**, **3c**, and **5h** showed the lowest values of IC_50_ in the ABTS test. With respect to the FRAP method, compound 3c has the strongest activity, followed by **5a**, **5h**, donepezil, **3m**, and **5j**. In the FTC method, the highest significant diminution was demonstrated by **3a** followed by **3c**, **5h**, and **5j**, compared with BHT. Thus, compound **3c**, a promising acetylcholinesterase inhibitor among the new derivatives, demonstrated very good antioxidant activity across the three tested methods (FRAP, ABTS, and FTC). Additionally, in vitro studies revealed that the melatonin derivative **3n** efficiently prevents oxidative stress-induced injury in SH-SY5Y cells, exhibiting the best BChE inhibitory activity. As a future perspective, we plan to evaluate hydrazones **3c**, **3d**, **3a**, **3n**, and **5a** in vivo using models of Alzheimer’s disease and melatonin deficiency, as well as Aβ (1–42) aggregation.

Importantly, the new series of melatonin derivatives containing the donepezil fragment showed low cytotoxicity and a good in vitro safety profile. Compounds **3a–d** (IC_50_ 129, 138.4, 97.3, 122.3 μM, respectively), **3k** (130.8 μM), and **5a** (142.0 μM) exhibited the best bioavailability, with IC_50_ values twice as high as the reference donepezil (79.3 ± 6.2 μM) in the human neuroblastoma cell line SH-SY5Y and three times higher in normal mouse fibroblasts. Notably, the most active compounds **3a–d** and **5a** displayed negligible cytotoxic activity in the mouse neuroblastoma cell culture Neuro-2a (IC_50_ > 300 μM); making them promising structures for further study in AD. An in silico pharmacokinetics analysis predicted that all tested hybrids could be well absorbed, metabolized, and excreted, with most of them capable of crossing the BBB. Additionally, the most promising compounds were measured by PAMPA for blood–brain barrier permeabilities.

Finally, docking studies suggest potential future applications of the MTDL **3c** in complex diseases such as major depression and AD, which involve targeting AChE and/or BChE enzymes and melatonin MT1 and MT2 receptors.

## Data Availability

Data is contained within the article and Appendix A.

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
