# Peer review of "Design, Synthesis, In Silico Studies and In Vitro Evaluation of New Indole- and/or Donepezil-like Hybrids as Multitarget-Directed Agents for Alzheimer’s Disease"

_pharmaceuticals, 2023, doi:10.3390/ph16091194_

Round 1

Reviewer 1 Report

The authors describe two novel series of melatonin- and donepezil-based hybrid molecules with hydrazine or sulfonyl hydrazone fragments. In particular, they were designed, synthesized and evaluated as multifunctional ligands, against AD-related neurodegenerative mechanisms. The work is interesting, but it is too dispersive in the presentation of the results. It is clear the methodology approach and results are well argued. In my opinion, the paper might be considered to be accepted after minor revisions.

Minor points

1. The nomenclature/numbering of the compounds is not clear. Initially, the two series are defined as 3* and 5*, then I found 3* and 6* and then again 3* and 6*. Better define and standardize.

2. In Tables 1 and 2 the calculation and the value of the compounds activity is not clear. In the tables they are indicated as a percentage value, but they are also defined in the reference at the end of the table as a concentration. Better define.

3. Figure 3 should be moved to Supplementary.

4. The docking results should highlight the difference of the binding mode between the two series of compounds.

5. Docking results towards MT1 and MT2 should be synthetized or partly moved to the supplementary.

6. A brief discussion of the results obtained with respect to the state of the art of the literature should be included in the manuscript.

Author Response

Dear Reviewer,

Thank you very much for the e-mail containing you're and  comments on the Manuscript ID: pharmaceuticals-2549678, entitled: Molecular docking, structure-based design and biological evaluation of a new indole- and/or donepezil-like hybrids as multitarget-directed agents for Alzheimer's disease. 

Enclosed, please, find attached our manuscript which has been revised and corrected following strictly all referees’ suggestions and annotations made in the PDF manuscript.

Sincerely,
Violina T. Angelova

Reviewer 2 Report

The manuscript entitled "Molecular docking, structure-based design and biological evaluation of a new indole- and/or donepezil-like hybrids as multi-target-directed agents for Alzheimer's disease'' reported the synthesis of thirty new benzylpiperidine and indole derivatives. The structure assignment of obtained products has been confirmed by NMR and Mass spectroscopy. AChE, BuChE inhibition and antioxidant activities of synthesized compounds were assessed. Results from the biological evaluation, in-silico molecular docking studies and ADMET analyses confirmed that, these compounds are promising prototypes that can be used to design potential lead compounds for the AD.
This manuscript is written in good English and has new ideas about multi-target-directed agents, but there are some errors that need to be modified.

Comments:

1.     Title: Structure-based design basically relies on the three-dimensional structure of the molecular target for drug. Therefore, title should be revised.

2.     Figure 1: Design strategy should clearly depict the structure of the newly designed molecule.

3.     Result and discussion: There should be a section related to the chemistry (including characterization) of the synthesized molecules.

4.     IR spectra of the synthesized molecules should be included in the characterization and supplementary.

5.     The authors should write elemental analysis for some compounds.

6.     Validation of docking protocol should be included.

7.     There are several typographical errors. Such as line# 119: in vitro, line#507: CH2 group, line#715: in vivo, etc.

This manuscript is written in good English 

Author Response

Dear Reviewer,

Thank you very much for the e-mail containing you’re comments on the Manuscript ID: pharmaceuticals-2549678, entitled: Molecular docking, structure-based design and biological evaluation of a new indole- and/or donepezil-like hybrids as multitarget-directed agents for Alzheimer's disease. Enclosed, please, find attached our manuscript which has been revised and corrected following strictly all referees’ suggestions and annotations made in the PDF manuscript.

Reviewer 3 Report

Dear Authors,

You present here a  research work regarding the investigation of some new indole- and/or donepezil-like hybrids as some new multi-target-directed agents for Alzheimer disease. The study of these new substance is very complex, you synthesized them, analyzed their structures, analyzed the safety profile and also the antioxidant activity. 

The references used are well-chosen, relevant for the theme and up to date.

There are still some suggestions and minor corrections on the manuscript:

1. write "in silico", "in vitro" in Italic

2. I observed that the number of some tables and figures are colored

3. on page 14, write H2O2 instead of H2O2

4. you should improve the clarity and quality of the figures from the molecular docking study. They are not very clear and are difficult to analyze

5. in some Figure legends and in page 32, lines 532-536, there seems to be a different font and letter size

6. in the characterization of compounds, their names are for some written in bold, for others non-bold

7. there are many studies done, but I consider that you should insist more on the correlation between them

8. you should detail the Conclusions part

Author Response

Dear Reviewer,
Thank you very much for the e-mail containing your commentson the Manuscript ID: pharmaceuticals-2549678, entitled: Molecular docking,structure-based design and biological evaluation of a new indole- and/or donepezil-likehybrids as multitarget-directed agents for Alzheimer's disease. Enclosed, please, find attached ourmanuscript which has been revised and corrected following strictly allreferees’ suggestions and annotations made in the PDF manuscript.  
Sincerely,
Violina T. Angelova  

Reviewer 4 Report

The manuscript by Angelova et al. describe synthesis and biological activity investigation of hybrid compounds containing two different pharmacophoric moieties based on donepezil and melatonin chemical structures. The antioxidant, in vitro, and in silico studies related to Alzheimer's disease were performed and discussed by the authors. The results are important and deserve publication. However, the manuscript contains numerous drawbacks listed below.

Line 29:  The abbreviation BChE is mentioned for the first time in Abstract and should be explained here (analogously to AChE).

Line 108:  It is desirable to highlight "in silico" in italic font (please check also throughout the manuscript).

Line 119:  It is desirable to highlight "in vitro" in italic font (please check also throughout the manuscript).

Lines 133-135:  The sentence concerning the separate ligand binding sites of AChE should be supplied with literature reference(s).

Lines 145-147:  The authors should clearly describe how the E-configuration was attributed to the synthesized compounds based on the NMR data. Also, it should be explained how the more abundant conformer (indicated in Materials and methods) at the amide bond was assigned.

Lines 139, 140:  It seems more suitable to replace "and" with "or".

Line 261:  The abbreviation BHT is mentioned for the first time in the manuscript and should be explained here.

Line 270:  The phrase "greenish radical" is incorrect from the chemical point of view. It is reasonable to write about the greenish solution of this radical.

Line 274:  Please replace "easily" with "easier".

Line 288:  The phrase "iron solubility" is incorrect in this context. It is reasonable to discuss the solubility of iron complexes.

Line 289:  The abbreviation "SET" should be explained here (single electron transfer?).

Table 4:  Compounds 6a-k are present in the table. However, their chemical structures are not given in the manuscript. The same refers to figures 4, 10, S2, and S8.

Line 337:  Please use subscripts in the hydrogen peroxide formula. Check also throughout the manuscript.

Line 366:  The enzyme is called "BuChE" here (and in some other paragraphs - check!), while the abbreviation "BChE" is mainly used  in other parts of the manuscript.

In the sections of the manuscript where the docking study is discussed, the computational results obtained for the synthesized compounds (series 3 and 5) are presented. Meanwhile, in Section "Chemistry and design strategy of the multifunctional donepezil-melatonin hybrids" (lines 142-144), the authors declare that the compounds were synthesized based on preliminary docking results, i.e., a pre-synthesis desing is implied. A detailed description of the principles of this docking-based design should be added somewhere in the manuscript.

Molecules 3e-i contain a chiral center (a carbon atom in the pyrrolidine ring). Its configuration is not indicated in Scheme 1 and in Materials and methods (proline-like?). A reader can understand this as though the compounds were synthesized and experimentally tested as racemic mixtures. Please clarify this explicitly. Moreover, docking results can be strongly dependent on a chosen enantiomer. It is not clear from the manuscript, which of the two enantiomers for each of these molecules was taken for the docking computations. These issues should be clearly addressed by the authors.

Figure 10:  The first two "negative" columns are not marked in the diagram.

Line 955:  The authors indicate that the E-isomer of compound 3r was synthesized. However, the corresponding Z-isomer is shown in Table 1.

Summarizing, I recommend major revision of the manuscript before acceptance.

Author Response

Dear Reviewer,
Thank you very much for the e-mail containing your comments on the Manuscript ID: pharmaceuticals-2549678, entitled: Molecular docking,structure-based design and biological evaluation of a new indole- and/or donepezil-likehybrids as multitarget-directed agents for Alzheimer's disease. Enclosed, please, find attached our manuscript which has been revised and corrected following strictly all referees’ suggestions and annotations made in the PDF manuscript.  
Sincerely,
Violina T. Angelova  

Reviewer 5 Report

Violina T. Angelova deposited paper entitled Molecular docking, structure-based design and biological evaluation of a new indole- and/or donepezil-like hybrids as multi- target-directed agents for Alzheimer's disease. Alzheimer's disease is a serious problem for people and a challenge for researchers. Authors consider two novel series of melatonin- and donepezil-based hybrid molecules with hydrazone or sulfonyl hydrazone (fragments as multifunctional ligands against neurodegenerative mechanisms in  Alzheimer's disease. They reveal two lead compounds exhibited a well-balanced multifunctional profile, demonstrating intriguing acetylcholinesterase inhibition, promising antioxidant activity and inhibition of lipid peroxidation in linoleic acid system. They suggest that compunds have low neurotoxicity against malignant neuroblastoma cell lines. Authors used diverse experimental and in silico methods to present reliable results. It seems reasonable. In silico predictions without experimental suport/proof would be unacceptable. In my opinion, the paper could be interesting for readers and stimulate further studies on effective therapeutics for Alzheimer`s disease. On the other hand, paper is unfinoshed from the technical side. For example: What does it mean     p–p aromatic stacking? (line 129) or cation–p interaction (line 131) Why numbering of schemes//figures are colored?   … and red fragments in the text „in the section below” (line 239)? Fig. 2 or Figure 2 should be mentioned in the text? Figures 3, 4, 6 and 7 shoudl be corrected – they are blurred.

Author Response

(The authors gave the same response as above.)

Round 2

Reviewer 4 Report

I am satisfied with the authors' responses and corrections in the manuscript. In my opinion, the manuscript can be accepted for publication in the revised form.

Reviewer 5 Report

Authors corrected paper. I have no more comments.